# Hydrogen sulfide aggravates neutrophil infiltration, vascular remodeling and elastase-induced abdominal aortic aneurysm in male mice

Clémence Bechelli[1], Diane Macabrey[1], Florian Caloz[1], Severine Urfer[1], Martine Lambelet[1], Florent Allagnat ®[1,2] ✉ & Sébastien Déglise[1,2]

## Abstract

**Background** Abdominal aortic aneurysm (AAA) has an 80% mortality rate upon rupture, with no pharmacological treatments available to slow its progression. Hydrogen sulfide ($H_2S$), produced by cystathionine γ-lyase (CSE), has anti-inflammatory and antioxidant properties, but its role in AAA remains unclear.
**Methods** We evaluated the impact of sodium thiosulfate (STS), a clinically relevant $H_2S$ donor, in a periadventitial elastase-induced AAA model in normotensive male wild-type and $Cse^{-/-}$ mice. Complementary in vitro studies were conducted on primary human vascular smooth muscle cells (VSMCs) to assess the effects of STS on proliferation, senescence and cytokine-induced apoptosis.
**Results** Contrary to expectations, STS dose-dependently aggravate AAA progression by increasing extracellular matrix degradation. Although STS reduces macrophage and lymphocyte infiltration, it enhances neutrophil accumulation, particularly MMP9+ neutrophils, and promotes the formation of c-KIT+-MPO+ pre-neutrophil clusters. $Cse^{-/-}$ mice show reduced neutrophil infiltration and smaller aneurysms, supporting a pathogenic role of endogenous $H_2S$. STS also impairs VSMC proliferation and induces senescence, blunting compensatory aortic remodeling.
**Conclusions** $H_2S$, delivered via STS, exacerbates AAA progression under normotensive conditions by promoting neutrophil-driven inflammation and impairing VSMC repair. These findings challenge the assumption that $H_2S$ is universally protective in vascular disease and raise caution regarding the therapeutic use of STS in patients at risk for AAA.

## Plain language summary

Abdominal aortic aneurysm (AAA) is a life-threatening condition where a major blood vessel in the abdomen, called the aorta, becomes weak and bulges. There are currently no medications that can slow down AAA growth, and rupture carries a high risk of death. Hydrogen sulfide ($H_2S$) is a gas naturally produced in the body, that has shown to protect against cardiovascular diseases. This study investigated whether sodium thiosulfate (STS), a $H_2S$-releasing compound, could reduce AAA progression in mice. Unexpectedly, STS worsened AAA. Our findings highlight the need for caution when considering STS as a treatment for patients at risk of AAA.

Abdominal aortic aneurysm (AAA) is a degenerative condition characterized by the localized dilation of the abdominal aorta, exceeding 50% of its normal diameter affecting 5% of males over 65 years of age[1,2]. AAA rupture has a devastating 80% death rate because most AAA are asymptomatic[3]. Current treatment options are limited to surgical interventions, such as open surgical repair or minimally invasive endovascular aortic repair (EVAR), both of which carry considerable risks[4]. There are no effective pharmacological treatments to halt or reverse AAA progression. This unmet medical need underscores the importance of understanding the molecular and cellular mechanisms driving AAA development.

While the precise mechanisms underlying the development of AAA are not yet fully elucidated, the pathogenesis of AAA was well characterized and described as multifaceted, involving chronic inflammation, oxidative stress, extracellular matrix (ECM) degradation, and vascular smooth muscle cell (VSMC) apoptosis[5]. Key features include the infiltration of innate and adaptive immune cells into the aortic wall, leading to persistent

[1]Department of Vascular Surgery, Lausanne University Hospital, Lausanne, Switzerland. [2]These authors jointly supervised this work: Florent Allagnat, Sébastien Déglise. ✉e-mail: Florent.Allagnat@chuv.ch

inflammation[5,6]. Neutrophils are among the earliest immune cells recruited, secreting proteolytic enzymes such as matrix metalloproteinases (e.g., MMP9) that degrade elastin and collagen, weakening the aortic wall. Macrophages also play a critical role by producing cytokines, chemokines, and additional proteases that sustain inflammation and matrix remodeling. While macrophages contribute to tissue damage, they also exhibit reparative functions, with M2-polarized macrophages promoting resolution of inflammation and tissue repair. Disruptions in the balance between these immune cell populations and their functions exacerbate ECM degradation and VSMC loss, driving AAA progression and eventual rupture.

A growing body of evidence also suggests that the process of VSMC senescence plays a pivotal role in the pathogenesis of AAA[7,8]. Smooth muscle cells are the primary structural components of the aortic wall, and their dysfunction or loss can contribute to the weakening and eventual dilation of the aorta. Cellular senescence, a state of irreversible cell cycle arrest, is increasingly recognized as a contributing factor in the development of various cardiovascular diseases, including AAA[7,8]. Specifically, senescent VSMCs feature properties of excessive degradation and disorganization of the ECM. Additionally, these senescent cells secrete pro-inflammatory mediators, exacerbating the local inflammatory milieu and perpetuating tissue damage. Importantly, the impairment of vascular repair mechanisms, such as the diminished capacity for smooth muscle cell proliferation and migration, hampers the aorta's ability to effectively remodel and regenerate in response to the ongoing insults, leading to progressive dilation and AAA formation[7,8]. Collectively, the multifaceted effects of VSMC senescence on the structure, function, and repair of the aortic wall may be pivotal drivers in the AAA pathogenesis.

Hydrogen sulfide (H₂S) is an endogenous gasotransmitter with well-documented antioxidant, anti-inflammatory, and vasorelaxant properties[9,10]. Produced enzymatically in mammalian cells through the transsulfuration pathway by cystathionine γ-lyase (CSE) and cystathionine β-synthase (CBS), H₂S plays critical roles in cardiovascular homeostasis[10]. Within the vasculature, H₂S is predominantly synthesized by CSE, with its activity linked to endothelial cell function and vascular smooth muscle tone[9,11–13]. Mice lacking Cse ($Cth^{-/-}$) on a mixed genetic background display age-dependent hypertension[14]. However, other strains of $Cth^{-/-}$ mice on a pure C57BL/6 genetic background are normotensive[15,16], suggesting an impact of the genetic background on this phenotype. Endothelial-specific $Cth^{-/-}$ mice are also normotensive but display impaired endothelial-mediated vascular relaxation[12,17]. Preclinical studies have demonstrated protective effects of H₂S in models of cardiovascular diseases, including atherosclerosis and myocardial ischemia[9]. However, its role in AAA remains unclear. While studies in hypertensive models suggest that H₂S donors mitigate aneurysm formation by reducing oxidative stress and vascular remodeling, these effects may be confounded by the antihypertensive properties of H₂S. Indeed, it was shown that the H₂S donor salt NaHS attenuates aortic remodeling in a model of aortic dissection induced by angiotensin II (Ang-II) infusion and β-aminopropionitrile fumarate (BAPN) treatment in WT mice[18]. Similarly, NaHS treatment rescued hypertensive 52 weeks old hypertensive $Cth^{-/-}$ mice from angiotensin II-induced aortic elastolysis and medial degeneration[19]. However in these studies, NaHS fully abolished Angiotensin II-induced hypertension and the role of H₂S in normotensive conditions and its direct impact on immune cell dynamics and vascular remodeling remains poorly understood.

To investigate the role of H₂S in AAA progression under normotensive conditions, we employed a mouse model of elastase-induced AAA combined with the administration of BAPN[20–22]. BAPN is a lysyl oxidase inhibitor that prevents cross-linking of elastin and collagen, leading to impaired recovery following elastase application and the formation of a chronic, growing AAA. To test the role of H₂S, we used sodium thiosulfate (Na₂S₂O₃; STS), a clinically relevant H₂S donor[16,23] and normotensive $Cth^{-/-}$ mice[16]. Our findings reveal that, contrary to expectations, H₂S treatment aggravates AAA progression by promoting neutrophil infiltration, matrix degradation, and impaired VSMC repair. These observations underscore the complex

and context-dependent effects of in vascular inflammation and remodeling, highlighting the need for careful evaluation of its therapeutic potential in AAA management.

## Methods

A list of models, reagents, chemicals, and equipment is included in the Supplementary Data 2.

### Mice

WT male mice (C57BL/6JRj mice) were purchased from Janvier Labs (Le Genest-Saint-Isle, France). Upon arrival, mice were acclimatized for 1 week minimum to the animal facility. Cth knockout (Cse−/−) mice were created by crossing Cth floxed mice (Cthtm1a(EUCOMM)Hmgu; RRID: IMSR_:Cmsu10294) with CMV-cre global cre-expressing mice (B6.C-Tg(CMV-cre)1Cgn/J)[16]. Genotyping was performed using mouse ear biopsies digested over night at 55 °C in DirectPCR lysis reagent (Cat:102-T; Qiagen) supplemented with proteinase K (Cat:1122470WT; Qiagen). Heterozygous and knockout mice were identified by PCR using the forward primer 5'-AGC ATG CTG AGG AAT TTG TGC-3' and reverse primer 5'-AGT CTG GGG TTG GAG GAA AAA-3' to detect the WT allele and the forward primer 5'-TTC AAC ATC AGC CGC TAC AG-3' to detect knockout allele using the GoTaq Green Master Mix (Cat: M782A; Promega), as previously[16]. Both mouse lines were on a pure C57BL/6 J genetic background. The line was subsequently maintained by breeding animals heterozygous for the mouse Cth null allele, regularly backcrossed against C57BL/6JRj mice obtained from Janvier Labs. All mice were housed in standard housing conditions with wood chip litter, house tubes, and nesting material (22 °C, 12 h light/dark cycle; lights on at 07:00 am), with ad libitum access to water and a regular diet (SAFE°150 SP-25 vegetal diet, SAFE diets, Augy, France). Mice (cages) were randomly treated or not with STS (Hänseler AG, Herisau, Switzerland) in the water bottle at 2.5 or 4 g/L to achieve 0.6 or 1 g/kg/day, changed three times a week. All experiments and data analysis were conducted in a blind manner using coded tags rather than actual group name. The study focused on male mice by design as male sex is a major risk factor for the development of AAA. This male bias makes male mice the most appropriate and relevant population for initial studies.

### AAA surgery

Abdominal aorta aneurysm surgery was performed under isoflurane anesthesia (2.5% v/v 2.5 L O₂). Analgesia was ensured by subcutaneous injection of buprenorphine (0.1 mg/kg Temgesic, Reckitt Benckiser AG, Switzerland) and local anesthesia via subcutaneous injection with a mix of lidocaine (6 mg/kg) and bupivacaine (2.5 mg/kg) along the incision line. 15 min post-injection, and while deeply anesthetized, a midline incision was made, and the aorta separated from the surrounding fascia below the kidneys. A Whatmann paper impregnated with 8 μL of pancreas porcine elastase solution (Cat: E1250-20mg; Merck KGaA, Darmstadt, Germany) was applied to the surface of the aorta and left in place for 10 min. Following Whatmann removal, the peritoneum cavity was rinsed with warm saline, the abdomen closed with Vicryl 6.0 sutures (Cat: W9981; Ethicon), and the skin closed with staples (Cat: 427631, Autoclip® 9 mm, CLAY ADAMS). Buprenorphine was provided before surgery, as well as a post-operative analgesic every 8 h for 36 h. The animals were monitored using a dedicated scoresheet for each mouse. Mice were observed twice daily for signs of distress (behavior, pain, activity, surgical wound) during the first 3 days post-op, then once a day the remaining experiment. Unless stated otherwise, all mice received BAPN (3-aminopropionitrile fumarate salt; Cat: A3134, Sigma-Aldrich,) via the drinking water at 0.2% w/w concentration from the day after the surgery until aorta collection. Mice (cages) were randomly (randomisation within blocks) assigned to the control or STS-treated groups post-surgery. Aortas were collected 14 days post-surgery at Zeitgeber time (ZT) 2-4 by cervical dislocation and exsanguination under isoflurane anesthesia, followed by PBS and 4% v/v buffered formaldehyde perfusion, fixed in buffered formalin for 24 h, and included in paraffin for histology studies.

## Systolic blood pressure measurements

Systolic blood pressure (SBP) was monitored daily by non-invasive plethysmography tail cuff method (BP-2000, Visitech Systems Inc.) on conscious mice[16,24] at ZT 6–9 after 1 week of procedural acclimatization.

## Histology

Aortas in paraffin were cut into 5 µm sections. Sections were stained with Verhoeff–van Gieson Elastic Lamina (VGEL) staining or Polychrome Herovici staining as described[25]. Young collagen was stained blue, while mature collagen was pink. Cytoplasm was counterstained yellow. Hematoxylin was used to counterstain nuclei blue to black. Images were acquired using a Zeiss AxioScan Z1 and proprietary ZEN Microscopy Software 3.9.

Histomorphometry analysis of the aortic wall was performed as follows: Aortas were isolated, and paraffin embedded. The whole sub-renal aorta was cut from the renal to the iliac arteries. 5 µm sections every 200 µm of the subrenal aorta were cut towards the iliac bifurcation. Cross sections were analyzed using the *Fiji (ImageJ 1.53t)* software. Aortic lumen area was extrapolated from the perimeter to correct for altered geometry of the fixed aortas. Perimeter was measured on 4 to 6 cross sections per aorta, separated by 200 µm. AAA size was expressed as the mean, max, and AUC of the AAA area from 4 to 6 AAA area measured along the length of the abdominal aorta. The max AAA area represented the largest AAA area, the mean was the average of 4 to 6 areas, and the AUC was derived from the AAA area curves over 1–1.5 mm of abdominal aorta. The elastin degradation grade was scored using VGEL or autofluorescence visualization of the elastic laminae on a scale of 0 to 4 grade, where 0 represents perfect elastic lamina as seen on native aorta, and 4 represents a complete loss of elastic laminae. Two independent researchers blind to the experimental groups did the morphometric measurements.

Immunohistochemistry was performed on paraffin sections. After rehydration and antigen retrieval (TRIS-EDTA buffer pH 9, 1 min in an electric pressure cooker (autocuiser Instant Pot duo 60 under high pressure), immunostaining was performed on abdominal aorta sections using the rabbit-specific HRP/DAB detection IHC detection kit (Cat: ab236469, Abcam) according to the manufacturer's instructions. The slides were further counterstained with hematoxylin. The antibodies used in that study are described in Supplementary Data 3. The positive immunostaining area was quantified in a semi-automatic manner by manual color thresholding using the *Fiji (ImageJ 1.53t)* software and normalized to the total area of the tissue by two independent observers blinded to the conditions.

## Fluorescent immunohistochemistry of mouse FFPE tissue (Akoya protocol)

Multiplex imaging of mouse FFPE tissue was performed using the Opal 3-Plex Anti-Rb Manual Detection Kit (Cat: NEL840001KT, Akoya Biosciences), according to the manufacturer's instructions using Opal 570 and Opal 690 secondary antibodies. Antigen retrieval was performed using TRIS-EDTA buffer pH 9, 1 min in an electric pressure cooker (autocuiser Instant Pot duo 60 under high pressure). For the NE or Ki67/MPO combination, slides were incubated with NE or Ki67 antibody over night at 4 °C, combined with Opal 570 incubation for 10 min at room temperature, followed by a second antigen retrieval and incubation with MPO at room temperature for 1 h combined with Opal 690 incubation for 10 min at room temperature. Slides were further washed and cooked to remove unbound fluorophores, counterstained with DAPI (Cat: H-1200, Vectashield) mounting medium, and images using a Zeiss AxioScan Z1 and proprietary ZEN Microscopy Software v. 3.9.

## Reverse transcription and quantitative polymerase chain reaction (RT-qPCR)

Grinded frozen gastrocnemius muscles or liver were homogenised in Tripure Isolation Reagent (Cat: 11667165001; Roche, Switzerland), and total RNA was extracted according to the manufacturer's instructions. After RNA Reverse transcription (Prime Script RT reagent, Cat: RR037B; Takara),

cDNA levels were measured by qPCR Fast SYBR™ Green Master Mix (Cat: 4385618, Applied Biosystems, ThermoFischer Scientific AG, Switzerland) in a Quant Studio 5 Real-Time PCR System (Applied Biosystems, ThermoFischer Scientific AG, Switzerland), using the primers given in the Supplementary Table 1.

## Western blot

Mouse aortas were flash-frozen in liquid nitrogen, grinded to powder, and resuspended in SDS lysis buffer (62.5 mM TRIS pH 6, 8, 5% SDS, 10 mM EDTA). Protein concentration was determined by a DC protein assay (Cat: 500-0116, Thermo Fischer Scientific AG, Switzerland). 10 to 20 µg of protein were loaded per well. Primary cells were washed once with ice-cold PBS and directly lysed with reducing Laemmli SDS sample buffer. Lysates were resolved by SDS-PAGE and transferred to Immobilon-P PVDF membranes (Cat: IPVH00010; Millipore), blocked in Tris Buffer Saline supplemented with 0.1 vol. % Tween 20 (TBS-T) and 5 wt. % powdered skimmed milk and for 1 h, and incubated overnight at 4 °C with the primary antibodies diluted in the same buffer. Blots were then washed 3 times 10 min in TBS-T, then incubated for 1 h at room temperature in HRP-conjugated antibodies diluted TBS-T-5 wt. % milk. Immunoblotting was performed using the antibodies described in the Supplementary Data 3. Membranes were revealed using Immobilon Western Chemiluminescent HRP Substate (Cat: WBKLS0050; Millipore) in an Azure Biosystems 280 (Azure Biosystems AG, Switzerland) and analyzed using Fiji (ImageJ 1.53t) software. Protein abundance was normalized to total protein using the PierceTM Reversible Protein Stain Kit for PVDF Membranes (Cat: 24585; Thermo Fischer Scientific AG, Switzerland).

## Flow cytometry

Peripheral blood was collected from the mice tail vein into EDTA tubes 3 days post-op. 50 µL of whole blood was combined with 50 µL of FACS buffer (Ca-Mg free PBS, 2 mM EDTA, 2% FCS) using the antibodies described in the Supplementary Data 3. The mixture was incubated for 30 min at room temperature. Then, without washing cells, 1 mL of Red Blood Cell lysing solution ($0.15$ M $NH_4Cl$, $5.7^{10-3}$ $KH_2PO_4$, $1^{10-4}$ $Na_2EDTA$) was added, mixed gently, and incubate for 5–8 min at room temperature. Leucocytes were then centrifuged at 400 g for 5 min and washed 3 times in 1 mL of FACS buffer at room temperature. Finally, cells were resuspended in 500 µL of FACS buffer for flow cytometry using a CytoFLEX flow cytometer (Beckman Coulter, Inc.). All analysis were conducted using the CytExpert Software version 2.4 (Beckman Coulter, Inc.) recording at least 10'000 events.

## Primary human VSMC culture

Human VSMCs were prepared from human saphenous vein segments as previously described[26,27]. Vein explants were plated on the dry surface of a cell culture plate coated with 1% gelatin type B (Sigma-Aldrich). Explants were maintained in DMEM/F-12, GlutaMAX™ medium (Cat: 31331-028; Thermo Fisher Scientific Inc.) supplemented with 5% FBS (Cat: FBS-16A, Capricorn Scientific), 1x Gibco™ Insulin-Transferrin-Selenium-Ethanolamine (Cat: 51500056; Gibco™), and 0.5 ng/mL Recombinant Human EGF (Cat: AF-100-15; PeproTech®, Thermo Fisher Scientific Inc.) and 2 µg/mL Recombinant Human bFGF (Cat: 100-18B; PeproTech®, Thermo Fisher Scientific Inc.). Cells were maintained in a cell culture incubator (Hera Cell; Thermo Fisher Scientific Inc.) at 37 °C in a 5% $CO_2$ and 5% $O_2$ environment. Nine different veins or patients were used in this study to generate VSMC. VSMC was used between passages 1 and 8.

## BrdU assay

VSMCs were grown at 80% confluence ($5 \cdot 10^3$ cells per well) on glass coverslips in a 24-well plate and starved overnight in serum-free medium. Then, VSMC were either treated or not (ctrl) with the drug of choice for 24 h in full medium in presence of 10 µM 5-bromo-2′-désoxyuridine (BrdU; Cat: B5002, Sigma-Aldrich). All conditions were tested in parallel and replicated

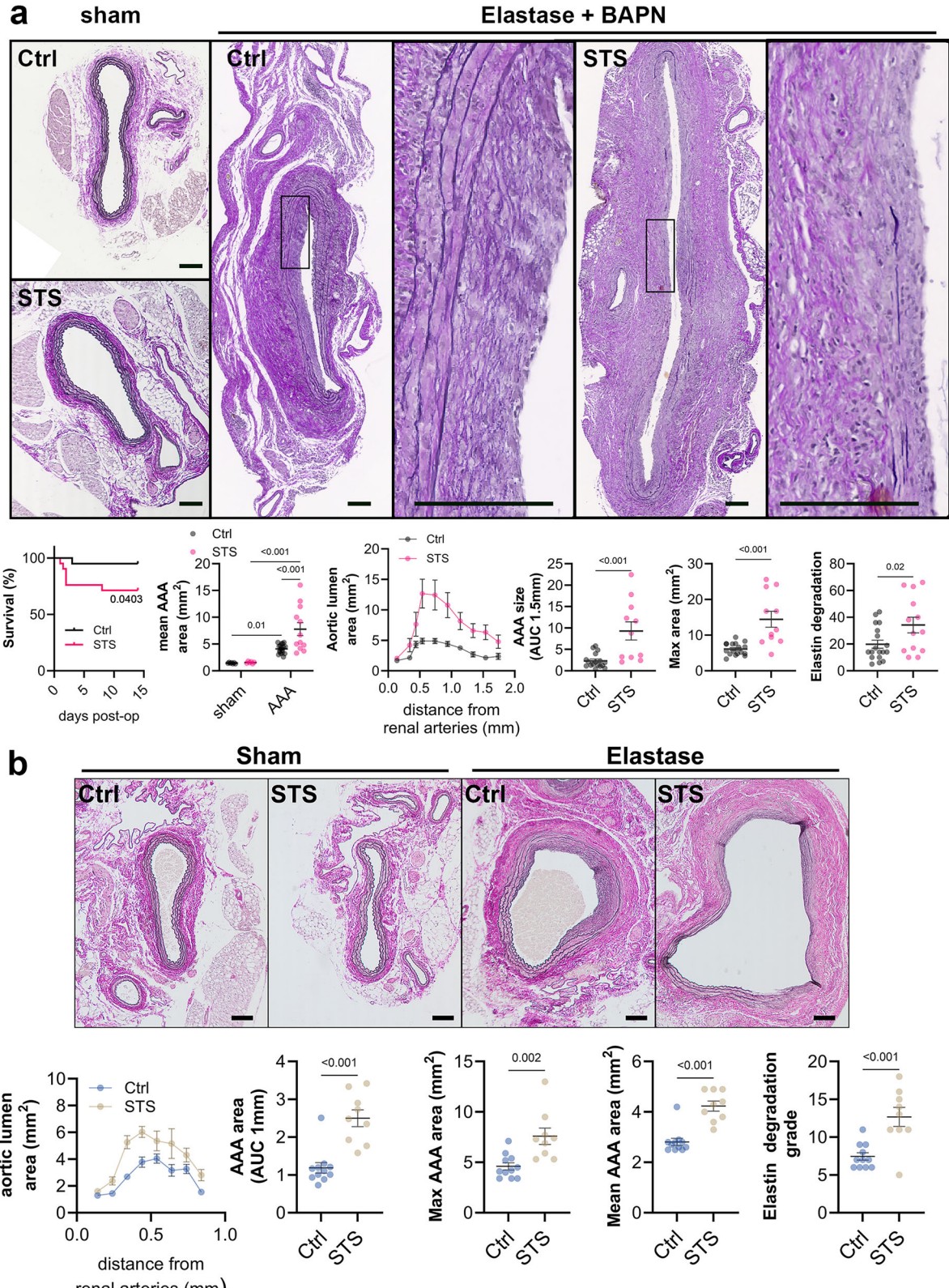

**Fig. 1 | STS increases AAA size in a mouse model of topical application of elastase.**
**a** Upper panels: Representative Verhoeff–van Gieson Elastic Lamina (VGEL) staining of native sub-renal aorta (sham) or 14 days post topical elastase application from male wild type (WT) mouse treated with β-aminopropionitrile (BAPN) and treated or not (Ctrl) with 4 g/L sodium thiosulfate (STS). Scale bar: 100 μm. Lower panels: Quantitative assessment of survival, mean lumen area in sham (7 Ctrl and 6 STS) and AAA (19 Ctrl and 12 STS-treated mice), aorta lumen area (curves and AUC over 1.5 cm), max lumen area, and elastin degradation grade. Data are mean ± SEM. **b** Upper panels: representative VGEL staining of native sub-renal aorta (sham) or 14 days post topical elastase application from male mouse treated or not (Ctrl) with 4 g/L STS (*no BAPN treatment*). Lower panels: quantitative assessment aorta lumen area (curves and AUC over 1 mm), max and mean lumen area, and elastin degradation grade. Scale bar: 100 μm Data are mean ± SEM of 11 Ctrl and 9 STS-treated mice. **a**, **b** *P*-values determined by bilateral unpaired t-test. Survival curves were compared by $X^2$ via Log-rank (Mantel-Cox) test.

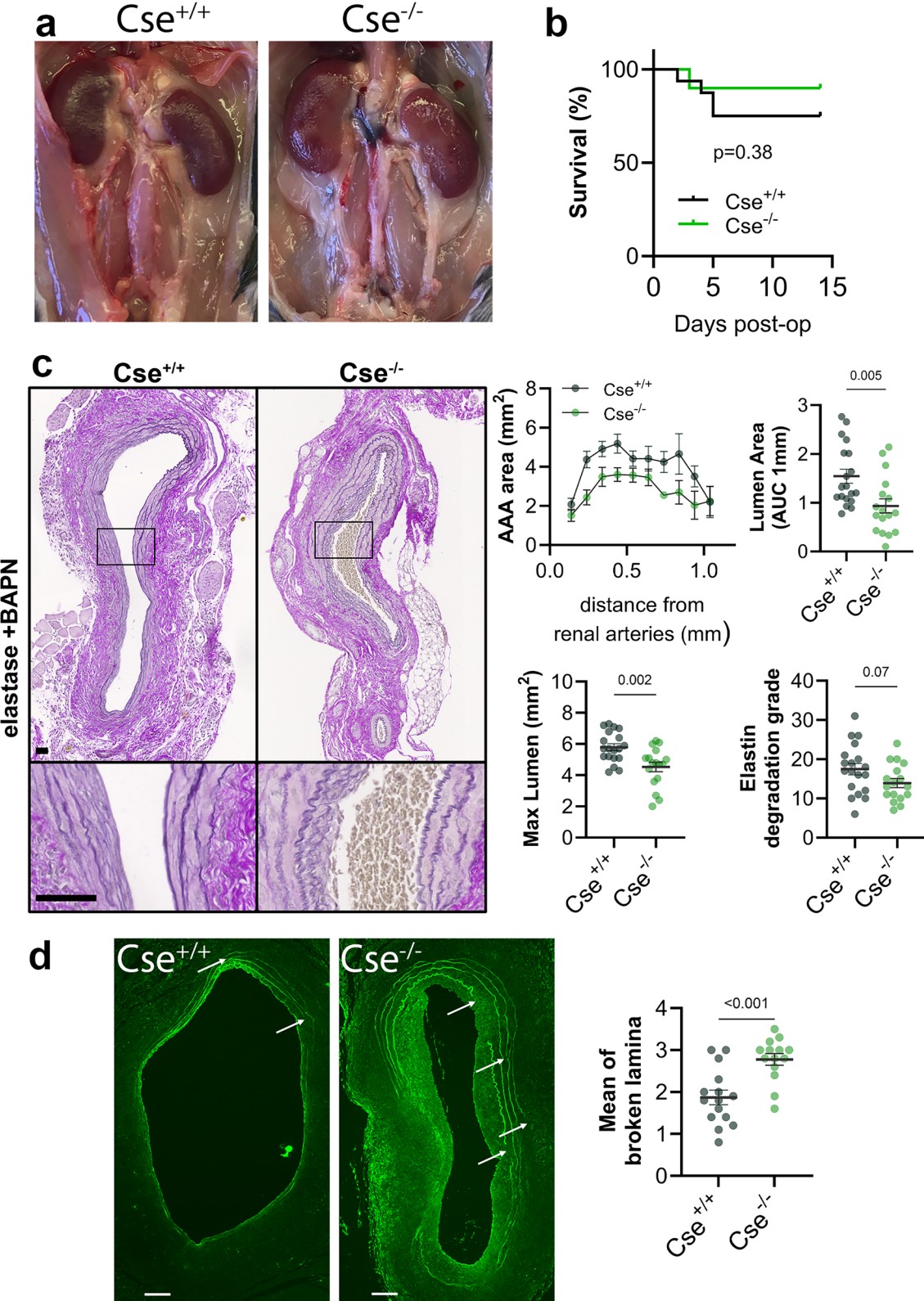

**Fig. 2 | Cse⁻/⁻ are protected against AAA.** Sub-renal AAA 14 days post-topical application of elastase with β-aminopropionitrile (BAPN) treatment in Cse⁻/⁻ and WT (Cse⁺/⁺) littermates. **a** Representative photograph of sub-renal AAA. Scale bar = 2 mm. **b** Kaplan-Meier survival curves of 16 Cse⁺/⁺ and 10 Cse⁻/⁻ mice. Survival curves were compared by X² via Log-rank (Mantel-Cox) test. **c** Representative Verhoeff–van Gieson Elastic Lamina (VGEL) staining (*left panels*) and quantitative assessment (*right panels*) of lumen area, AAA lumen area (area under the curve (AUC) over 1 cm), max lumen area, and elastin degradation grade. Scale bars represent 80 μm. Data are mean ± SEM of 20 Cse⁺/⁺ and 17 Cse⁻/⁻ mice. *P* values determined by bilateral unpaired t-test. **d** Representative autofluorescence (*left panels*) and quantitative assessment (*right panels*) of elastin breaks in 15 Cse⁺/⁺ and 14 Cse⁻/⁻ mice. White arrows point to elastin breaks. *P*-values determined by bilateral unpaired t-test.

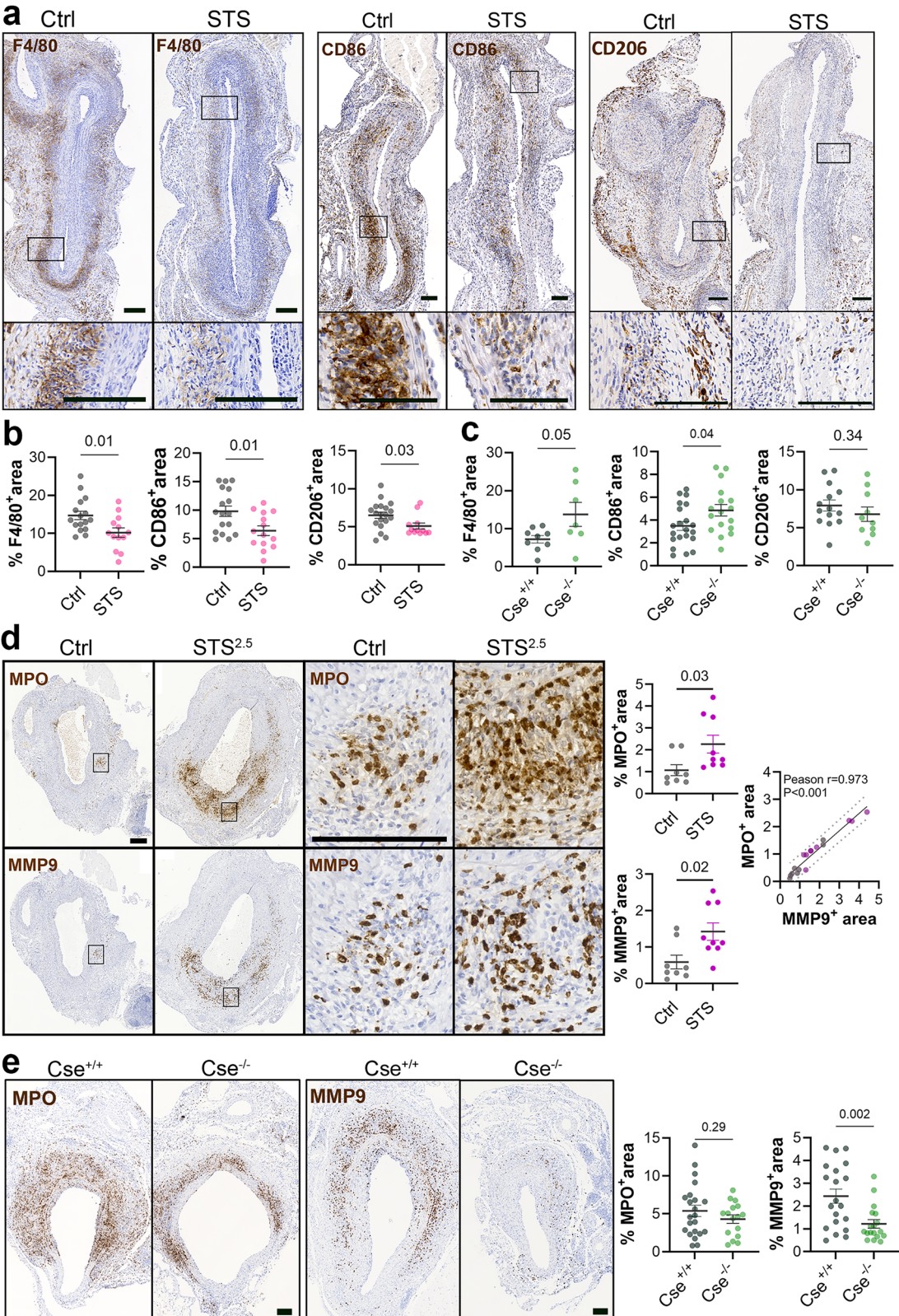

as independent experiments. All cells were fixed in ice-cold methanol 100% after 24 h of incubation and immunostained for BrdU. Images were acquired using a Nikon Eclipse 90i microscope using the NIS Element software v6.10.01. BrdU-positive nuclei and total DAPI-positive nuclei were automatically detected using the Fiji *(ImageJ 1.53t)* software[27].

### RNA interference

CSE knockdown was performed using human siRNA targeting CTH (Ambion-Life Technologies, ID: s3710 and s3712). The control siRNA (siCtrl) was the AllStars Negative Control siRNA (Qiagen, SI03650318). VSMC grown at 70% confluence were transfected overnight with 30 nM

**Fig. 3 | H$_2$S promotes neutrophil infiltration in the AAA wall.** Representative images **a** and quantitative assessment **b** of F4/80, CD86, and CD206 immunostaining and in sub-renal aorta in WT male mice 14 days post topical elastase application, treated or not (Ctrl) with 4 g/L sodium thiosulfate (STS), and with β-aminopropionitrile (BAPN). Scale bars are 100 μm. Data are mean ± SEM of 16 Ctrl and 14 STS-treated mice. P values determined by bilateral unpaired t-test. **c** Quantitative assessment of F4/80, CD86 and CD206 infiltration in sub-renal aorta in Cse$^{+/+}$ or Cse$^{-/-}$ male mice 14 days post topical elastase application. Data are mean ± SEM of 9 to 19 Cse$^{+/+}$ and 7 to 17 Cse$^{-/-}$ mice. *P* values determined by bilateral unpaired t-test. **d** Representative myeloperoxidase (MPO) and matrix metalloproteinase-9 (MMP9) immunostaining (left panels) and quantitative assessment (*right panels*) of MPO and MMP9 positive areas, and MPO-MMP9 Pearson correlations in sub-renal AAA in WT mice treated or not (Ctrl) with 2.5 g/L STS (STS$^{2.5}$). Scale bars are 100 μm. Data are mean ± SEM of 8 Ctrl and 9 STS-treated mice. **e** Representative MPO and MMP9 immunostaining (left panels) and quantitative assessment (right panels) of MPO and MMP9 positive areas in sub-renal AAA in Cse$^{+/+}$ and Cse$^{-/-}$ mice as indicated. Scale bars are 100 μm. Data are mean ± SEM from 20 Cse$^{+/+}$ and 16 Cse$^{-/-}$ mice. P values determined by bilateral unpaired t-test and Pearson correlation coefficient.

siRNA using lipofectamin RNAiMax (Cat: 13778–075, Invitrogen). After washing, cells were maintained in full media for 48 h prior to assessment.

### Seahorse
Mitochondrial stress tests were performed on confluent HUVECs according to the manufacturer's kits and protocols (Cat: 103015-100; Seahorse XF glycolysis stress test kit, Agilent Technologies, Inc.) in a XFe96/XF96 Analyzer (Agilent Technologies, Inc.). 1 μM Oligomycin was used. Data were analyzed using the Seahorse Wave Desktop Software (Agilent Technologies, Inc., Seahorse Bioscience).

### Mitochondrial staining
The mitochondrial network was observed by live cell imaging using the Mitotracker Red CM-H$_2$XRos fluorescent probe (Cat: M7513, Thermo-Fischer). The probe was dissolved in anhydrous DMF at 1 mM and used at 1 μM in serum-free RPMI. Live-cell image acquisition was performed using a Nikon Ti2 spinning disk confocal microscope. Images were analyzed automatically using the MiNA (Mitochondrial Network Analysis) toolset[28] in the Fiji (ImageJ 1.53t) software.

### Apoptosis and caspase 3/7 activity
VSMC were grown on a 96-well plate. At the end of the incubation period, the nuclei of cells were stained using 5 μg/mL propidium iodide (Cat: P3566, Sigma-Aldrich) and 5 μg/mL Hoechst 33342 (Cat: H1399, Sigma-Aldrich) diluted in cell culture medium. After a 15 min incubation in a cell culture incubator (37 °C in a 5% CO2 and 5% O$_2$ environment), the percentage of apoptotic cells was determined by manual counting of live and apoptotic and/or necrotic nuclei identified by their morphology after DNA staining as previously described[29]. The cells were examined by inverted fluorescence microscopy (Leica). A minimum of 500 cells was counted in each experimental condition by two independent observers, one of whom was unaware of the sample identity.

Caspase 3/7 activity was measured using the Apo-ONE® Homogenous Caspase 3/7 Assay (Cat: G7790; Promega). VSMC were grown on a 96-well plate, and 50 μl of the reagent was added to 50 μl of medium in each well. Blanks are composed of the reagent and medium. After one hour, fluorescence (excitation: 485 ± 20 nm; emission: 530 ± 20 nm) was detected in a multimode plate reader (Synergy H1, Biotek AG).

### Ethics statement
Human great saphenous vein segments were obtained from donors who underwent lower limb bypass surgery. All vein samples used for VSMC primary cultures were discarded tissue from patients who signed the institutional informed consent of the Lausanne University Hospital for the use of discarded biological material. VSMC cultures were derived from both male and female donors. However, as required by law, samples were fully anonymized upon collection, including age and sex, and no sex-based analysis could be performed. The study protocols for tissue collection and use were reviewed and approved by the Centre Hospitalier Universitaire Vaudois (CHUV) and the Cantonal Human Research Ethics Committee (http://www.cer-vd.ch/, no IRB number, Protocol Number 170/02), and are in accordance with the principles outlined in the Declaration of Helsinki of 1975, as revised in 1983 for the use of human tissues.

All animal experiments conformed to *The National Research Council: Guide for the Care and Use of Laboratory Animals*[30]. All animal experiments (design, group allocation, procedures, treatments) were approved and pre-registered by the Cantonal Veterinary Office. All animal care, surgery, and euthanasia procedures were approved by the Cantonal Veterinary Office (SCAV-EXPANIM, authorization number 3703). The study focused on male mice by design as male sex is a major risk factors for the development of AAA. This male bias makes male mice the most appropriate and relevant population for initial studies.

### Statistics and reproducibility
All experiments adhered to the ARRIVE guidelines and followed strict randomisation within blocks. All experiments and data analysis were conducted in a blind manner, using coded tags rather than actual group names. A power analysis was performed prior to the study to estimate the sample size. Based on previous experience, using a detectable difference of 40% in aorta diameter by histomorphometry, a standard deviation of 20%, a desired power (1-β) of 0.8, and a p value of 0.05 (alpha = 0.05), it was determined that a total of 10–12 animals in each group is necessary to reach statistically meaningful conclusions. In the in vivo experiment, animals are used as the experimental unit, so that each data point is an individual animal (n refers to number of animals per group). For in vitro studies the replicate unit is one experiment including all conditions in parallel with appropriate controls. Several independent experiments were conducted across few weeks using several batches/passages of cells to ensure reproducibility and minimize batch effects. All experiments were analyzed using GraphPad Prism 10.4. The normal distribution of the data was assessed using Kolmogorov-Smirnov tests. All data with a normal distribution were analyzed by bilateral Student's t-tests or mixed-effects model (REML) analysis, followed by post-hoc t-tests with the appropriate correction for multiple comparisons. For non-normally distributed data, Kruskal-Wallis non-parametric ranking tests were performed, followed by Dunn's multiple comparisons test to calculate adjusted *p*-values. Pearson's r correlation analyses were performed using GraphPad Prism 10.4. Unless otherwise specified, *p*-values are reported according to the APA 7th edition statistical guidelines. *$p < 0.05$, **$p < 0.01$, ***$p < 0.001$. The images presented in the manuscript are representative of the data and the image/staining quality.

## Results
### STS promoted aneurysm growth in WT mice in the model of elastase-induced AAA
We first assessed whether STS treatment protected against AAA growth and rupture in the model of periadventitial elastase-induced aortic wall digestion in WT mice treated with BAPN. As compared to native aorta (sham), periadventitial elastase combined with BAPN treatment led to the formation of large sub-renal AAA (Fig. 1a, Supplementary Fig. 1) with elastin degradation beyond the targeted area of the aortic wall, indicative of impaired wall repair and AAA progression. Surprisingly, STS treatment (4 g/L) increased the AAA size (Fig. 1), as quantified by the area under the curve of the aortic lumen area, and the mean and max lumen area (Fig. 1a). STS treatment also reduced survival, suggesting that STS led to aortic rupture. STS also increased elastolysis in WT mice (Fig. 1a). Of note, STS treatment had no impact on lumen area in native aorta (sham; Fig. 1a).

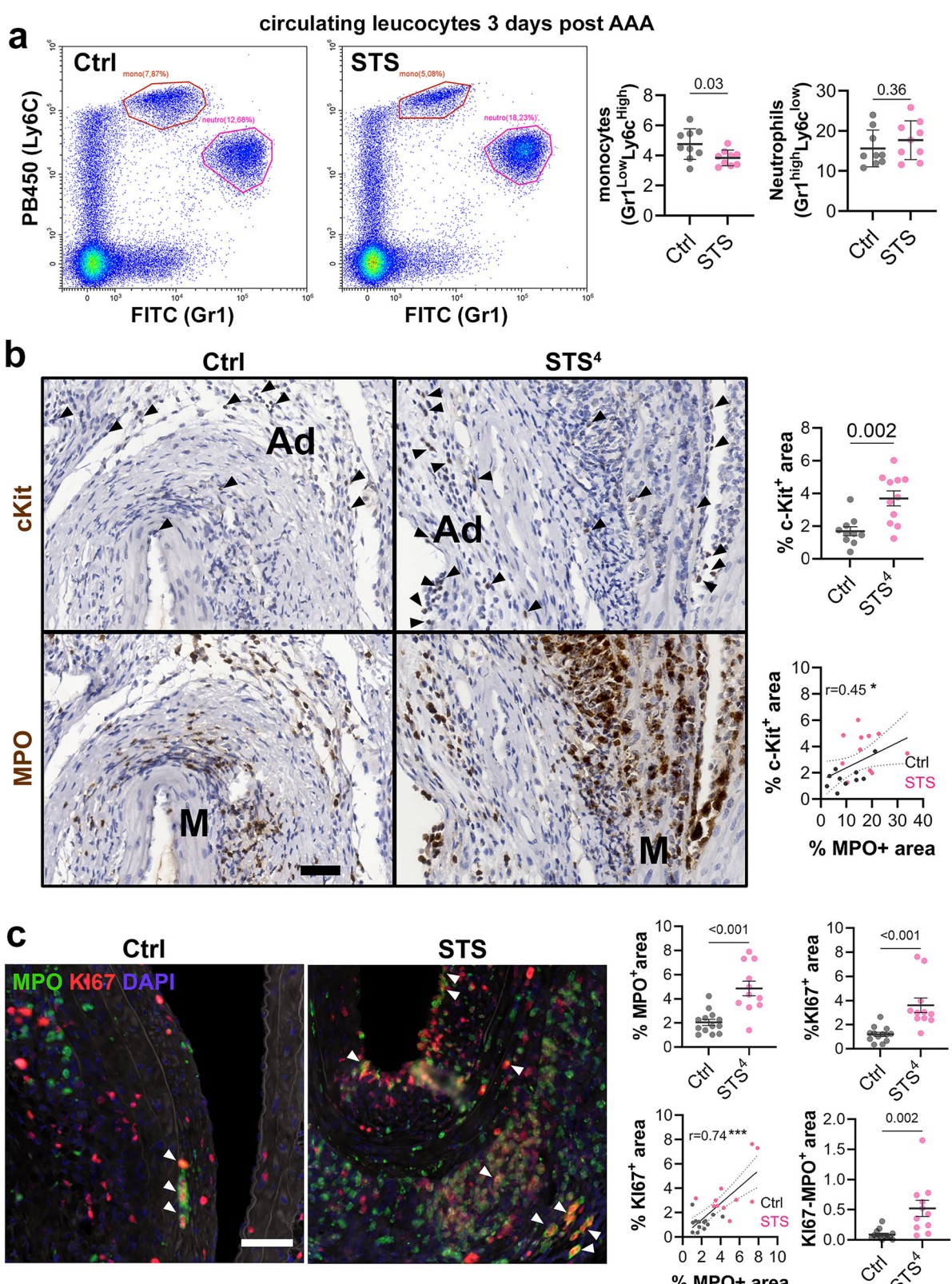

To assert that STS did not increase AAA size due to a toxic effect of $H_2S$, we performed a similar experiment using 2.5 g/L STS, which was recently shown to promote vascular repair[23]. 2.5 g/L of STS still promoted AAA size and elastolysis, but to a lesser extent than 4 g/L (Supplementary Fig. 2), suggesting a dose-dependent deleterious effect of STS on AAA growth. To test that the deleterious impact of STS was not due to interaction with the BAPN treatment, we also perform the same experiment as in Fig. 1A without BAPN treatment. AAA size and elastin degradation were reduced by 50% in absence of BAPN, indicative of ongoing elastic lamina repair in absence of BAPN (Fig. 1b). STS treatment still produced larger AAA using this model of periadventitial elastase application alone (Fig. 1b).

**Fig. 4 | H₂S promotes neutrophil proliferation in the AAA wall. a** Representative flow cytometry antibody staining and gating strategies (left panels; $10^3$ events in leucocyte gate recorded) for identification of monocyte and granulocytes in peripheral blood from WT male 3 days post-AAA surgery. Quantitative assessment (*right panels*) of gated monocyte and neutrophil populations in control mice (Ctrl) or mice treated with 4 g/L sodium thiosulfate (STS) and with β-aminopropionitrile (BAPN). Data are mean ± SD of 9 animals per group. *P* values as determined by bilateral unpaired t-test. **b** Representative immunostaining (left panels) and quantitative assessment (right panels) of c-KiT and Myeloperoxidase (MPO) positive signal in sub-renal aorta in WT male mice 14 days post topical elastase application, treated or not (Ctrl) with 4 g/L STS (STS), and with BAPN. Scale bars are 50 μm. Black arrowheads point to c-KiT⁺ cells. Ad adventitia, M media. Data are mean ± SEM of 10 Ctrl and 11 STS-treated mice. **c** Representative images (*left panels*) and quantitative assessment (right panels) of MPO and KI67 double immunostaining in sub-renal aorta WT male mice 14 days post topical elastase application, treated or not (Ctrl) with 4 g/L STS (STS⁴) and BAPN. Scale bars are 50 μm. White arrowheads point to MPO/KI67⁺ cells. Data are mean ± SEM of 13 Ctrl and 11 STS-treated mice. *P* values determined by bilateral unpaired t-test and Pearson correlation coefficient.

CSE is the main enzyme responsible for endogenous H₂S production in the vascular system, and Cse⁻/⁻ mice have impaired H₂S production capacity[16]. The Cse⁻/⁻ mice developed by Prof Rui Wang display age-dependent hypertension[14]. Here, we used normotensive Cse⁻/⁻ developed by late Prof. James R. Mitchell[16]. We confirmed that Cse⁻/⁻ mice did not express *mouse Cth*, and that there were no compensatory conges in H₂S-producing related *mouse* genes *Cbs* and *Mpst* (Supplementary Fig. 3a). Cse⁻/⁻ also displayed normal SBP between 8 and up to 65 weeks old (Supplementary Fig. 3b) as previously described[16]. When submitted to the periadventitial elastase-induced AAA with BAPN treatment model, male Cse⁻/⁻ mice developed smaller AAA than their WT littermates (Cse⁺/⁺) (Fig. 2a-c), with comparable survival rates (Fig. 2b). However, despite smaller AAA, Cse⁻/⁻ mice displayed an increased incidence of elastin breaks (Fig. 2d) and 22% of aortic dissection in Cse⁻/⁻ mice, whereas no aortic dissection was observed in Cse⁺/⁺ mice.

### STS reduced macrophages and lymphocytes infiltration but promoted MMP9⁺ neutrophil infiltration in the aortic wall

Inflammatory cells play a major role in AAA expansion[6]. Using the elastase + BAPN treatment model, we observed massive infiltration of MPO⁺ neutrophils and F4/80⁺ macrophages in the aortic wall 14 days post-surgery. To study the impact of STS on inflammation, we quantified immune cell infiltration in the AAA wall. In mice treated with 4 g/L STS (STS⁴), the infiltration of F4/80⁺ macrophages and CD86⁺ antigen-presenting cells (APCs) was reduced by 30 to 40%, as well as CD206⁺ anti-inflammatory M2 macrophages (Fig. 3a-b). On the contrary, Cse⁻/⁻ mice displayed increased infiltration of F4/80⁺ macrophages and CD86⁺ antigen-presenting cells, but not CD206⁺ macrophages (Fig. 3c). STS treatment in WT mice also reduced the infiltration of CD3⁺ and CD8⁺ lymphocytes (Supplementary Fig. 4a), whereas lymphocyte infiltration was increased in Cse⁻/⁻ mice (Supplementary Fig. 4b). In contrast, STS treatment (2.5gr/L STS, *p* = 0.03, Fig. 3d; 4 gr/L STS-P = 0.07, Supplementary Fig. 5) increased the infiltration of MPO⁺ neutrophils, while Cse⁻/⁻ tended to reduce MPO⁺ neutrophils infiltration (Fig. 3e). Further co-immunostaining for MPO and neutrophil elastase (NE) confirmed increased neutrophil infiltration in the aortic wall of WT mice treated with 4g/L STS (Supplementary Fig. 5). MMP9 is one the major protease secreted by neutrophils involved in AAA formation and aortic dissection[31,32]. STS treatment promoted the infiltration of MMP9⁺ cells, which correlated with MPO⁺ cells, suggesting neutrophils as the main source of MMP9 in that model (Fig. 3d-e and Supplementary Fig. 6a). In contrast, MMP9⁺ neutrophils infiltration was reduced in Cse⁻/⁻ mice (Fig. 3e). IL-6 is a major cytokine secreted by neutrophils[33]. STS also promoted IL-6 expression in the aortic wall (Supplementary Fig. 6b). However, neutrophils were not the only source of IL-6 as its signal also decorated the endothelium and media layer of the aorta.

To better understand the effect of STS on neutrophils and macrophages, we performed a flow cytometry analysis of circulating monocytes and granulocytes 3 days post AAA surgery (Supplementary Fig. 6c). STS decreased circulating Gr1^Low-Ly6c^High monocytes, suggesting decreased mobilization from the spleen and bone marrow (Fig. 4a) In contrast, STS did not impact circulating Gr1^High-Ly6c^Low granulocytes/neutrophils (Fig. 4a). Neutrophils are terminally differentiated cells that lack the ability to divide. However, recent studies, including a model of elastase-induced AAA, have identified a subpopulation of proliferative c-KiT⁺ pre-neutrophils in chronic inflammation[34,35]. Here, we observed the presence of c-KiT⁺ cells in the aortic wall (Fig. 4b). Consecutive slides staining with MPO highlighted that these c-KiT⁺ cells may be neutrophils, with a significant correlation between MPO and c-KiT⁺ cells (Fig. 4b). Of note, the c-KiT⁺ cells were detected mostly in the adventitia layer of AAA in Ctrl mice. The c-KiT⁺ population was increased in STS-treated mice, especially in the intima and media layer (Fig. 4b). To confirm the presence of proliferative pre-neutrophils in the AAA wall, double immunostaining for MPO and Ki67 was performed, revealing the existence of discreet clusters of double positive MPO/Ki67 cells (Fig. 4c). STS-treated mice displayed increased cell proliferation and double positive MPO/Ki67 cells, with a significant correlation between MPO and Ki67⁺ cells (Fig. 4c).

### STS inhibited respiration, promoted VSMC cell cycle arrest, and early senescence

In AAA, loss of elastic lamellae and matrix remodeling leads to reprogramming and proliferation of quiescent VSMCs. This VSMC expansion results in the formation of a neointima layer that can stabilize the aortic wall, providing additional support upon elastin degradation. Careful assessment of media and neointima thickness in the elastase-induced AAA model confirmed compensatory VMSC expansion in the control condition (Fig. 5a). STS treatment induced thinning of the media and neointima layer, suggesting decreased VSMC expansion (Fig. 5a). In contrast, Cse⁻/⁻ mice displayed increased media and neointima thicknesses compared to Cse⁺/⁺ mice (Figs. 2d, 5c). Further immunostaining of the VSMC-specific marker Calponin⁺ and MYH11⁺ revealed that both 2.5 and 4 g/L STS treatment significantly reduced Calponin⁺ and MYH11⁺ VSMC coverage (Fig. 5b, Supplementary Fig. 7a). However, Calponin⁺ and MYH11⁺ VSMC coverage did not change in Cse⁻/⁻ mice (Fig. 5c). Of note, STS did not seem to impact the identity of VSMC as measured by the expression of VSMC markers Calponin, SMA, and SM22α expression in native aortas, suggesting no direct effect of H₂S on VSMC identity (Supplementary Figs. 7b and 8). STS dose-dependently inhibited the proliferation of primary *human* VSMC in vitro (Fig. 6a). Furthermore, high throughput nuclei imaging indicated that cells exposed to STS for 24 or 48 h displayed larger nuclei (Supplementary Fig. 9). This change in cell morphology is reminiscent of cells undergoing senescence[7,8]. A 24 h exposure to STS dose-dependently increased the expression of cell cycle markers Cyclin B1, and markers of cell senescence p21^Cip1/Waf1 and γH2AX (Fig. 6b and Supplementary Fig. 10). STS also dose-dependently increased the markers of cell senescence p16^INK4A (Supplementary Figs. 11a, 12a) and the stress kinase JNK (Supplementary Figs. 11b, 12b). A 48 h exposure to STS stimulated AMPK phosphorylation, suggesting ATP depletion, and further stimulated the senescence marker p21^Cip1/Waf1, as well as the senescence-associated pro-inflammatory cytokine IL-1β (not detectable at 24 h) (Fig. 6c and Supplementary Fig. 13). We further confirmed that a 24 h exposure to 10 mM STS increased the proportion of p21-expressing nuclei, which correlated with the size of the nuclei (Fig. 6d). As expected for a H₂S donor, a 4 h STS treatment inhibited mitochondrial respiration and reduced ATP production in VSMC (Fig. 6e). A 24 h STS treatment also negatively impacted the mitochondrial morphology as assessed by live Mitotracker staining and determination of the mitochondrial network (Fig. 6f).

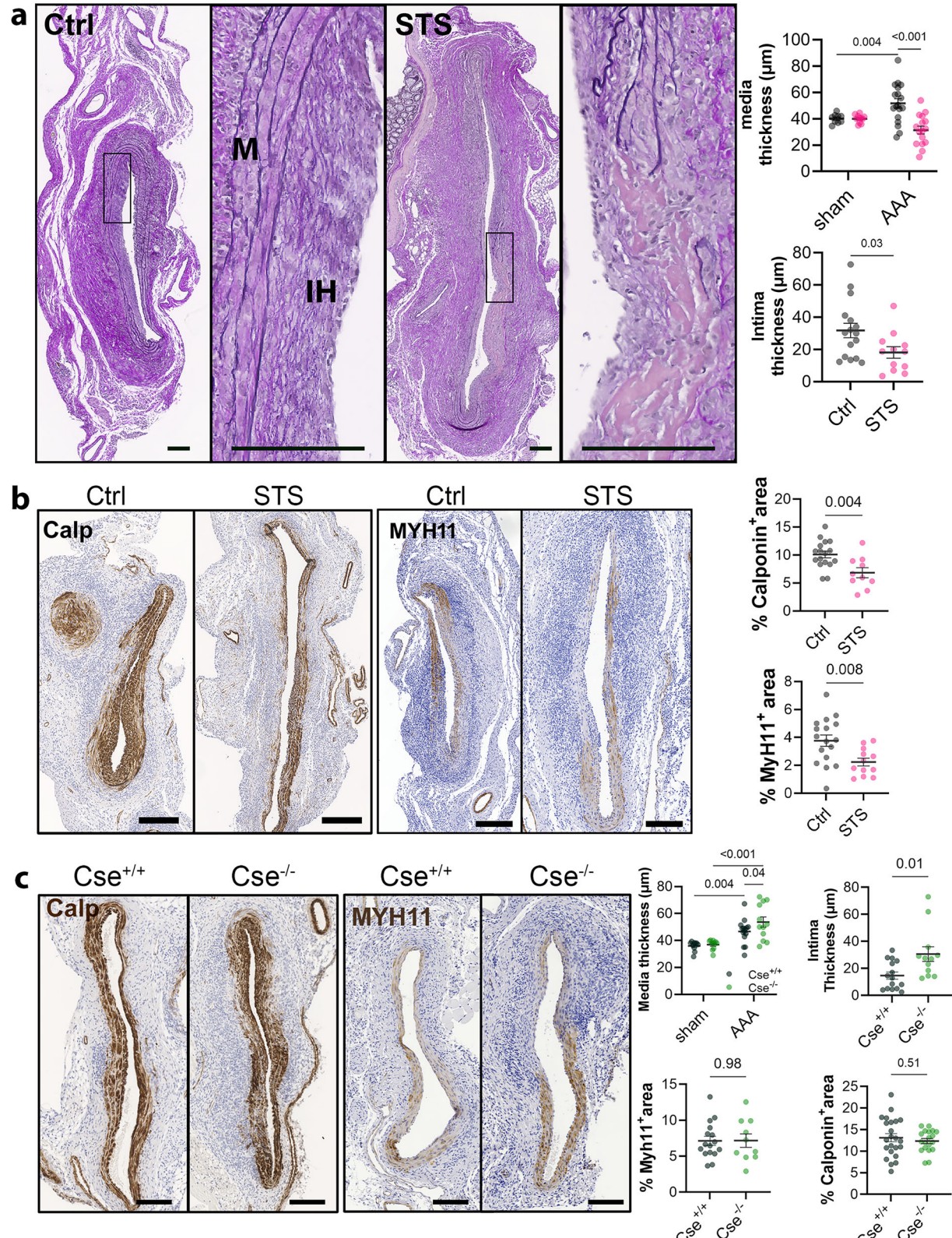

**Fig. 5 | STS reduces VSMC coverage in AAA. a** Representative VGEL staining (left panels) and assessment (right panels) of media and neointima thickness in sub-renal aorta in native aorta (sham; 8 animals per group) or 14 days post topical elastase application in 17 Ctrl and 12 sodium thiosulfate (STS)-treated mice. Scale bars are 100 µm. Data are mean ± SEM of 9–17 animals per group. P values as determined by matched mixed effect model analysis followed by post-hoc t-tests with Tukey's correction for multiple comparisons, or by bilateral unpaired t-test (intima thickness). **b** Calponin and Myosin-11 (MYH11) representative immunostaining (left panels) and quantitative assessments (*right panels*) of positive area in the sub-renal aorta 14 days post topical elastase application in 17 Ctrl vs. 12 STS-treated mice. **c** Calponin and MYH11 representative immunostaining (left panels) and quantitative assessments (right panels) of media and neointima thickness in sub-renal aorta in native aorta (sham; 8 animals per group) or 14 days post topical elastase application in 16 Cse[+/+] and 10 Cse[−/−] mice. Scale bars are 100 µm. *P*-values as determined by bilateral unpaired t-test or two-way ANOVA (media thickness) with Bonferroni correction for multiple comparisons.

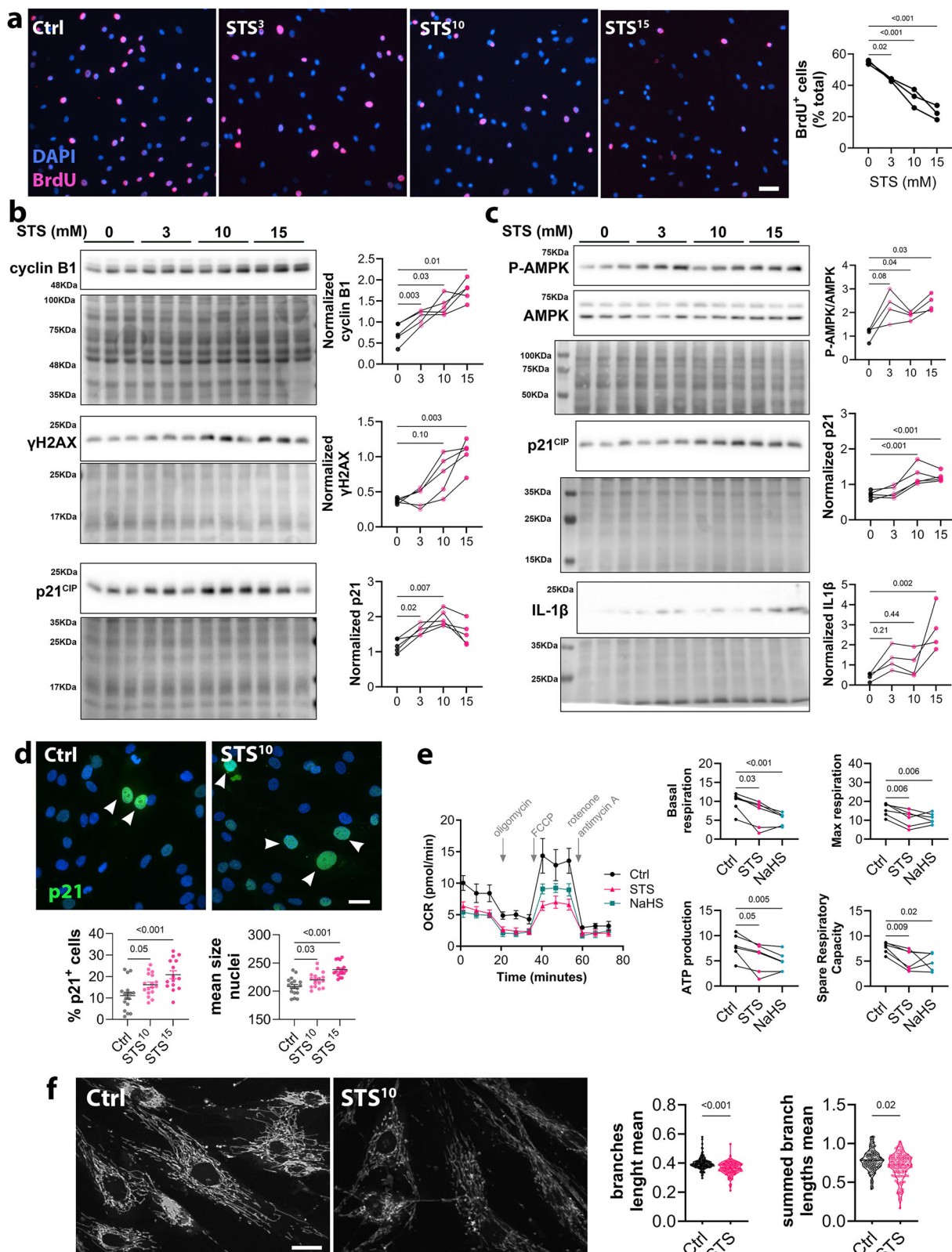

## STS aggravates cytokine-induced VSMC senescence and apoptosis

STS promoted neutrophils infiltration in the aortic wall, so we further tested the impact of STS on *human* VSMC in vitro in presence of only 10 ng/mL IL-6, and 10 ng/mL TNFα, two major pro-inflammatory cytokines secreted by neutrophils. Interestingly, IL-6 plus TNFα, but not IL-6 or TNFα alone,

also stimulated the appearance of large nuclei after 24 h (Fig. 7a) or 48 h (Fig. 7b) of incubation, suggesting that this cytokine combination also promoted VSMC senescence, but to a lesser extent than STS alone. The combination of STS with these cytokines further aggravated the number of senescent large nuclei (Fig. 7a, b) and the expression of senescence markers p21 (Fig. 7c and Supplementary Fig. 14a). Furthermore, in mice treated with

**Fig. 6 | STS accelerates replicative senescence in primary human VSMC. a** Cell proliferation (BrdU incorporation) in vascular smooth muscle cells (VSMC) treated or not (Ctrl; 0 mM) with sodium thiosulfate (STS), as indicated. P values as determined by repeated measures (matched) mixed-effects model (REML) with followed by Dunnett's multiple comparisons tests in 3 independent experiments. Scale bars are 50 μm. **b, c** Representative Western blot analysis of Cyclin B1, γH2AX, p21$^{CIP}$, P-AMPK, AMPK, and IL-1β in VSMC treated for 24 h (**b**) or 48 h (**c**) with increasing concentration of STS, as indicated. Representative blots (left panels) and quantitative assessment (right panels) of protein expression, relative to total protein content, in 4 to 5 independent experiments. P values as determined by repeated measures mixed-effects model (REML) with followed by Dunnett's multiple comparisons tests. **d**Upper panels: Representative P21$^{CIP}$ immunocytochemistry (green) and nuclei (DAPI; blue) in VSMC treated for 24 h with 10 mM STS. Lower panels: quantitative assessment of P21$^+$ cells and nuclei size. Data are pooled scatter plots with mean ± SEM of 4 independent experiments. *P* values as determined by one-way ANOVA followed by Dunnett's multiple comparisons tests. **e** Mitochondrial stress test assay (Seahorse) in VSMC pre-treated or not (Ctrl) for 4 h with 100 μM NaHS or 10 mM STS. Representative traces (left panel) and quantitative assessment (right panels) in 6 independent experiments. *P*-values as determined by repeated measures mixed-effects model (REML) with followed by Dunnett's multiple comparisons tests. **f** Representative images and quantitative assessment of live Mitotracker staining in VSMC treated for 24 h with 10 mM STS. Data are pooled scatter plots of five independent experiments. *P*-values as determined by Kolmogorov-Smirnov test.

4 g/L STS (STS$^4$), the number of P16$^{INK4A+}$ nuclei increased in the aortic wall of AAA WT male mice, especially in dedifferentiated VSMC in the neointima media (M) and neointima (NI) regions (Fig. 7d), suggesting that STS also promoted VSMC senescence in vivo. siRNA-mediated CSE knockdown also increased the percentage of p21$^+$ large nuclei (Fig. 7e) and the expression of senescence markers p21 as assessed by WB (Fig. 7f and Supplementary Fig. 14b) or immunocytochemistry (Fig. 7g).

Cytokine-induced VSMC apoptosis contributes to AAA progression and rupture[36]. We could not evaluate the impact of STS on apoptosis in vivo due to very low numbers of cleaved caspase 3$^+$ cells in *mouse* AAA samples (Supplementary Fig. 15). Here, a 48 h exposure to IL-6, TNFα, or IL-6 plus TNFα did not induce cleaved caspase 3/7 activity (Fig. 8a) and apoptosis (Fig. 8b) in primary *human* VSMC. STS, alone or in combination with IL-6 and TNFα, also had no impact on cell apoptosis (Fig. 8a, b). However, IL-6 plus TNFα stimulated the expression of pro-apoptotic BCL2 proteins BAX, BID and BIM (Fig. 8c and Supplementary Fig. 16a), and STS tended to aggravate the effect of IL-6 plus TNFα on BID and BIM expression (Fig. 8c and Supplementary Fig. 16a). STS stimulated IL-1β expression by VSMC (Fig. 6d). Thus, VSMC were further treated with a cocktail of pro-inflammatory cytokines composed of 1 ng/mL IL-1β and 10 ng/mL TNFα, or 1 ng/mL IL-1β, 10 ng/mL TNFα and 10 ng/mL IL-6. The addition of IL-1β to the cytokine mix resulted in VSMC apoptosis and STS promoted this cytokine-induced cleaved caspase 3/7 activity (Fig. 8d) and apoptosis (Fig. 8e), tended to aggravate the effect of the three cytokines on BIM, BID and BAX (Fig. 8f and Supplementary Fig. 16b). In contrast, siRNA-mediated CTH knockdown protected against cytokine-induced VSMC cleaved caspase 3/7 activity (Fig. 8g) and apoptosis (Fig. 8h), and reduced cytokine-induced BIM and BAX, but not BID expression (Fig. 8i and Supplementary Fig. 16c).

## Discussion

Given the anti-inflammatory and antioxidant properties of H$_2$S[9,37], we hypothesized that STS, a clinically relevant source of H$_2$S[16,23], would protect against aneurysm growth in a mouse model of elastase-induced AAA. Surprisingly, STS dose-dependently promoted AAA growth and rupture, whereas Cse$^{-/-}$ mice were protected.

AAA's primary pathogenic features are i) infiltration of innate and adaptive immune cells in the aorta wall; ii) proteolysis of the extracellular matrix (ECM); and iii) loss of VSMC[5,8,38,39]. Here, H$_2$S inhibited the infiltration of macrophages and T-cells, but increased the accumulation of neutrophils promoting ECM degradation. STS also reduced beneficial VSMC expansion to stabilize the aortic wall upon elastin degradation, and facilitated cytokine-induced VSMC senescence and apoptosis, leading to increased AAA growth and rupture.

Inflammation is a key hallmark of AAA pathology[5,38]. H$_2$S is a potent anti-inflammatory molecule in the context of cardiovascular diseases[9,10]. Here, we observed as expected that H$_2$S reduced the infiltration of both innate and adaptive immune cells, including macrophages, CD86$^+$ APC, and lymphocytes. However, this did not correlate with a reduced AAA size. It could be that the anti-inflammatory effect of H$_2$S on macrophages is detrimental to AAA growth as it also reduces the infiltration of CD206$^+$ M2 macrophages, which have been shown to promote tissue repair[38,40]. Other immune cell populations, such as CD4$^+$ regulatory T cells[5,38], might be similarly impacted, leading to AAA growth. However, it is more likely that the deleterious effect of H$_2$S is linked to neutrophil infiltration, which largely contributes to the deterioration of the aortic wall in AAA[41]. Indeed, H$_2$S promoted the accumulation of MMP9$^+$ neutrophils in the aortic wall. This result is in contradiction with studies showing that chemotaxis and recruitment of polymorphonuclear cells (PMN) are inhibited by the H$_2$S donor sodium sulfide[42–46]. However, it is in agreement with studies showing that H$_2$S promotes PMN adhesion, migration, and survival in various models of acute inflammation and sepsis[47–51]. Interestingly, it was recently shown using Cse$^{-/-}$ mice that CSE promotes an excessive innate immune response while suppressing the adaptive immune response in response to Mtb infection[52], which is similar to our findings that Cse$^{-/-}$ display decreased neutrophil accumulation but increased T-cell infiltration. To better understand the role of Cse and H$_2$S, Further studies should be performed to carefully assess the expression of CSE and other H$_2$S-producing enzymes (CBS, 3MST) in the immune cell infiltrating the aortic wall and the vascular cell types (SMC, EC, and fibroblasts) of AAA versus native aortas.

Neutrophils are terminally differentiated cells that lack the ability to divide. Neutrophils also have a short lifespan and rapidly undergo apoptosis and phagocytosis, which is important for the resolution of inflammation[41,53]. However, recent studies, including a model of elastase-induced AAA, have identified a subpopulation of proliferative c-Kit$^+$ pre-neutrophils in chronic inflammation[34,35]. Here, we confirmed the presence of c-Kit$^+$ and Ki67$^+$ neutrophils in the aortic wall, with MPO/Ki67 double-positive clusters. Notably, the c-Kit$^+$ and Ki67$^+$ cells are concentrated in the adventitia, indicating the formation of a distinct niche of immature neutrophil progenitors at the periphery of the aortic tissue. This aligns with emerging evidence of maladaptive responses to chronic sterile inflammation, where committed proliferative neutrophil precursors arise within inflamed tissue[35]. Further phenotypic and functional characterization is needed to elucidate their origin, role in aneurysm pathology, and potential as therapeutic targets. In addition, H$_2$S promoted the accumulation of MMP9$^+$ neutrophils and IL-6 secretion. This suggests that H$_2$S may directly impact the phenotype/activity of neutrophils, which is in line with early studies showing that sulfite acts as an agonist of neutrophils[54,55]. A recent study comparing several H$_2$S donors with distinct kinetics of H$_2$S release highlighted that most donors promote neutrophil survival and oxidative burst[56]. Of note, it was also shown that activated neutrophils produce sulfite, and that neutrophil oxidative burst may drive polysulfide overload in the presence of excessive H$_2$S[57,58], which may contribute to the neutrophil's arsenal in fighting infections. This supports the concept that endogenous H$_2$S directly contributes as an immune cell effector. Overall, our data support that H$_2$S has both pro and anti-inflammatory effects on the immune system, which explains why H$_2$S has been reported to promote or inhibit inflammation depending on the experimental models and approaches. While our data suggest that H$_2$S promotes the local expansion of neutrophil to the AAA wall, further studies are required to elucidate how H$_2$S influence the phenotypic signatures of distinct neutrophil subtypes, and their potential interactions with other immune cells (macrophages, T cells, B cells, dentritic cells).

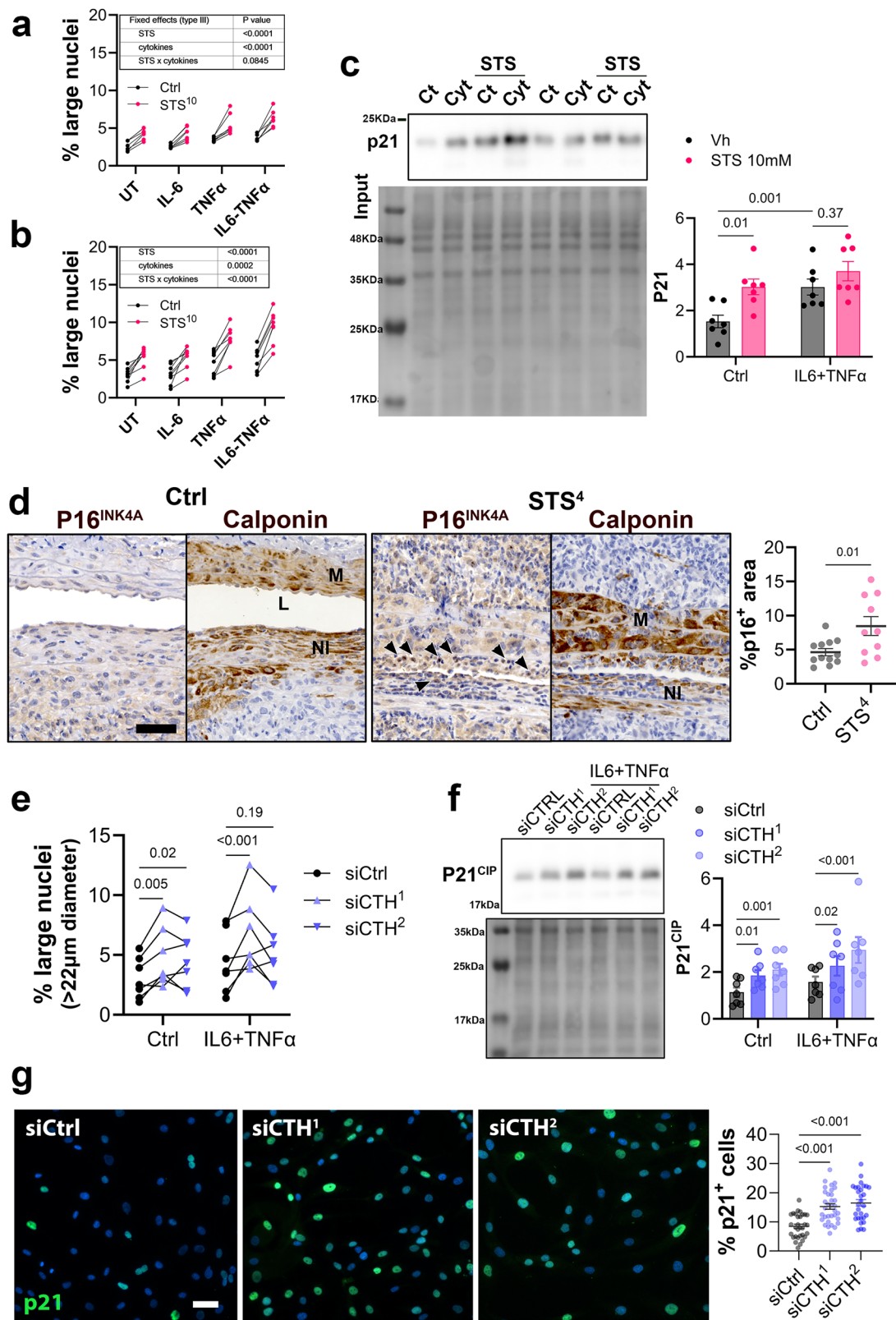

Besides its effect on immune cells, we observed that $H_2S$ inhibits VSMC proliferation as previously shown[16,27,59]. Inhibition of VSMC proliferation in our model likely led to reduced Calponin[+] and MYH11[+] cell coverage and media thickness, contributing to the weakening of the aortic wall. At the molecular level, we further observed that STS induced early VSMC replicative senescence as evidenced by altered nuclei morphology, and

expression of senescence marker γH2AX, P21[CIP], and P16[INK4A], and senescence associated secretory phenotype via the production of IL-1β. The presence of senescent VSMC is a known feature of advanced aneurysm lesion[7,8]. Interestingly, while STS treatment promoted VSMC senescence, reduced $H_2S$ production following CSE knock-down also promoted VSMC senescence. It is not surprising that CSE knock-down promoted senescence

**Fig. 7 | Both STS and CSE knock-down promote VSMC senescence.** High throughput live measurement of nuclei size (Hoechst staining) in vascular smooth muscle cells (VSMC) treated or not (untreated; UT) for 24 h (**a**) or 48 h (**b**) with 10 mM sodium thiosulfate (STS) and interleukin-6 (IL-6) and/or tumor necrosis factor alpha (TNFα), as indicated. Data are the mean ± SEM of 6 or 7 paired independent experiments. *P* values as determined by matched (paired) mixed-effects model (REML). **c** Representative Western blot (left panels) and quantitative assessment (right panel) of p21$^{CIP}$ in VSMC treated or not (Ctrl) for 24 h with STS and/or IL-6 + TNFα. Data are mean ± SEM of 7 paired independent experiments. *P* values as determined by matched (paired) mixed-effects model (REML), followed by Tukey's multiple comparisons tests. **d** Representative P16$^{INK4A}$ and Calponin immunostaining (*left panels*), and quantitative assessments of P16$^{INK4A}$ positive area (right panel) in the sub-renal aorta 14 days post topical elastase application in 12 Ctrl

and 10 STS (4 g/L)-treated mice. Scale bar represents 50 μm. Black arrowhead point to P16$^{INK4A}$ positive cells in the neointima layer. P value as determined by bilateral unpaired t-test. **e, g** Live measurement of nuclei size (**e**), p21$^{CIP}$ protein expression (**f**), and percentage of p21$^+$ cells (**g**; p21$^{CIP}$ immunocytochemistry (green) and nuclei (DAPI; blue)) in VSMC knocked-down for cystathionine gamma-lyase (CTH gene) using 2 distinct siRNAs (siCTH $^1$ and $^2$) and treated or not (UT) for 24 h with IL-6 + TNFα. **e, f** Data are the mean ± SEM of 7 paired independent experiments. *P*-values as determined by matched (paired) mixed-effects model (REML), followed by Dunnett's multiple comparisons tests. **g** Pooled data from 4 independent experiments, shown as scatter plots with mean ± SEM. *P*-values as determined by matched (paired) mixed-effects model (REML), followed by Dunnett's multiple comparisons tests.

as it was recently shown that CSE sequesters p53 in the cytosol to prevent its nuclear translocation and cell cycle arrest in old endothelial cells[60]. Of note, this role of CSE seems independent of H$_2$S production[60]. Further studies are required to clearly understand how STS promotes senescence, but it could be related to inhibition of mitochondrial respiration. Indeed, the deleterious effects of H$_2$S have often been attributed to inhibition of mitochondrial cytochrome C oxidase (Complex IV)[61], and mitochondrial dysfunction is a feature of senescence[62,63]. Here, we observed that STS rapidly inhibits mitochondrial ATP production, leading to altered mitochondrial network and AMPK activation. We propose that the effect of STS on VSMC proliferation plays a substantial role in AAA growth by preventing VSMC expansion to stabilize the weakened aortic wall in the early phase of AAA formation[40]. This hypothesis is in line with evidence that the VSMC clonal expansion takes place early in aneurysm development in both human aneurysm samples and mouse models[64]. Interestingly, despite pre-clinical studies showing that anti-inflammatory medications prevent AAA formation and/or dissection, anti-inflammatory medications in humans have failed[65,66], which has been proposed to be due to the cytostatic effect of immunosuppressive medications on VSMC[65,67].

Finally, H$_2$S increases cytokine-induced VSMC apoptosis. VSMC apoptosis is a hallmark of aortic aneurysm progression and rupture[36]. Our analysis of cleaved caspase 3 in the aortic wall did not allow us to evaluate apoptosis in vivo. This is probably due to the rapid phagocytosis of dying cells by immune cells in vivo. Our findings revealed that STS alone is not cytotoxic, as previously demonstrated[16], but facilitates cytokine-induced VSMC apoptosis, while CSE knock-down protects against cytokines. Elevated levels of exogenous H$_2$S may cause cell cycle arrest and apoptotic cell death[68]. Similarly to our findings, H$_2$S has been previously shown to promote ROS-induced mitochondrial apoptosis via the Bax/Bcl2 pathway[68,69]. Here, H$_2$S-induced cell cycle arrest probably tips the balance between pro- and anti-apoptotic signals toward apoptosis, notably via regulation of BAX, BIM, and BID. These findings support that new avenues of research should focus on preventing VSMC apoptosis rather than reducing inflammation.

The study focuses on male mice by design as male sex is a major risk factor for the development of AAA. This male bias makes male mice the most appropriate and relevant population for initial studies. While it is true that women also develop AAAs, they typically appear after menopause. However, mice do not undergo menopause, making it difficult to model female AAA pathophysiology in a preclinical setting. Including pre-menopausal female mice in the study may yield irrelevant or misleading results and does not align with the 3Rs principles of ethical animal research. That said, it would be important and relevant to further study the impact of H$_2$S donors in a relevant model of female mice.

It could be argued that the deleterious impact of STS on AAA growth is due to a "toxic" level of H$_2$S. However, STS treatment was already deleterious at 0.6 g/Kg/day, a concentration shown to promote angiogenesis and vascular repair[23], and it was previously shown that STS is not harmful at concentrations up to 2 g/Kg/day[16,70–72]. Thus, the detrimental effect of STS on AAA growth cannot be attributed solely to the dosage employed in our study. While there was no noticeable increase in VSMC coverage in Cse$^{-/-}$ mice, loss of Cse improved positive wall remodeling. The effect of H$_2$S on

neutrophils was also evident in Cse$^{-/-}$ mice, indicating that endogenous levels of H$_2$S already promoted accumulation of MMP9$^+$ neutrophils. That said, the effect of STS on VSMC senescence is probably due to harmful levels of H$_2$S, even though the used concentration remains below the threshold to trigger VSMC apoptosis. Of note, VSMC senescence was observed in primary cells from venous origin, which may behave differently from aortic VSMCs. Further studies would be useful to confirm our findings in arterial VSMCs.

Our results are in contradiction with studies reporting beneficial effects of CSE and H$_2$S donors against the formation of aortic aneurysm and dissection[18,19,73]. Importantly, these studies employed models of angiotensin II-induced hypertension, leading to aortic dissection and aneurysm. H$_2$S is a commonly known vasodilator, and H$_2$S donors, including STS, lower blood pressure in various models of hypertension[74,75], including angiotensin II-induced hypertension[70,76]. Therefore, the vasoactive properties of Cse/H$_2$S protect against aortic dissection in hypertension-induced AAA models. Similarly, NaHS treatment normalizes blood pressure in hypertensive Cse$^{-/-}$ mice, leading to reduced angiotensin II-induced aortic elastolysis and medial degeneration[19]. Of note, the Cse$^{-/-}$ mice used in our study are normotensive[16], and all our experiments were performed in normotensive conditions. That said, in line with previous studies, we observed increased elastin breaks in Cse$^{-/-}$ mice, leading to aortic dissection. This aortic wall dissection did not lead to aortic rupture but may indicate increased aortic stiffness related to impaired vasoreactivity. Indeed, CSE deficiency may alter the balance between structural integrity and functional response to hemodynamic forces. Thus, the presence of increased elastin breaks in the Cse$^{-/-}$ mice may reflect a failure in adaptive remodeling mechanisms that normally protect the aorta from mechanical stress during AAA formation. A lack of H$_2$S-induced vasorelaxation might exacerbate the mechanical stress on the aortic wall, leading to the formation of elastin breaks and localized dissection, all the while the overall aneurysm size remains smaller due to the lack of excessive neutrophil infiltration and matrix degradation. H$_2$S has been shown to influence the production of collagen and elastin by vascular tissues[77,78]. While Cse$^{-/-}$ mice show less matrix degradation (likely due to reduced neutrophil-drive MMP9 activation), we cannot exclude impaired matrix synthesis and repair, particularly VSMC derived matrix synthesis in Cse$^{-/-}$ mice. Conversely, the vasorelaxant property of STS may ease AAA growth in the elastase-induced AAA model. It should be noted that we were not able to directly measure the AAA size in vivo, and we acknowledge that luminal area measurements from cross-sections may be influenced by tissue handling and mounting artifacts, which could introduce some variability in AAA size quantification. Moreover, although 100% survival was observed in control mice, we cannot fully exclude the possibility that mortality in the STS-treated group resulted from causes other than aneurysm rupture.

Although the elastase-induced AAA model is regarded as the best model for human AAA disease[79–81], it does not reproduce the natural history of human AAA. Further research employing different AAA models, such as the CaCl$_2$ model[79,80], might be beneficial to confirm and better understand the role of H$_2$S in AAA formation in normotensive conditions.

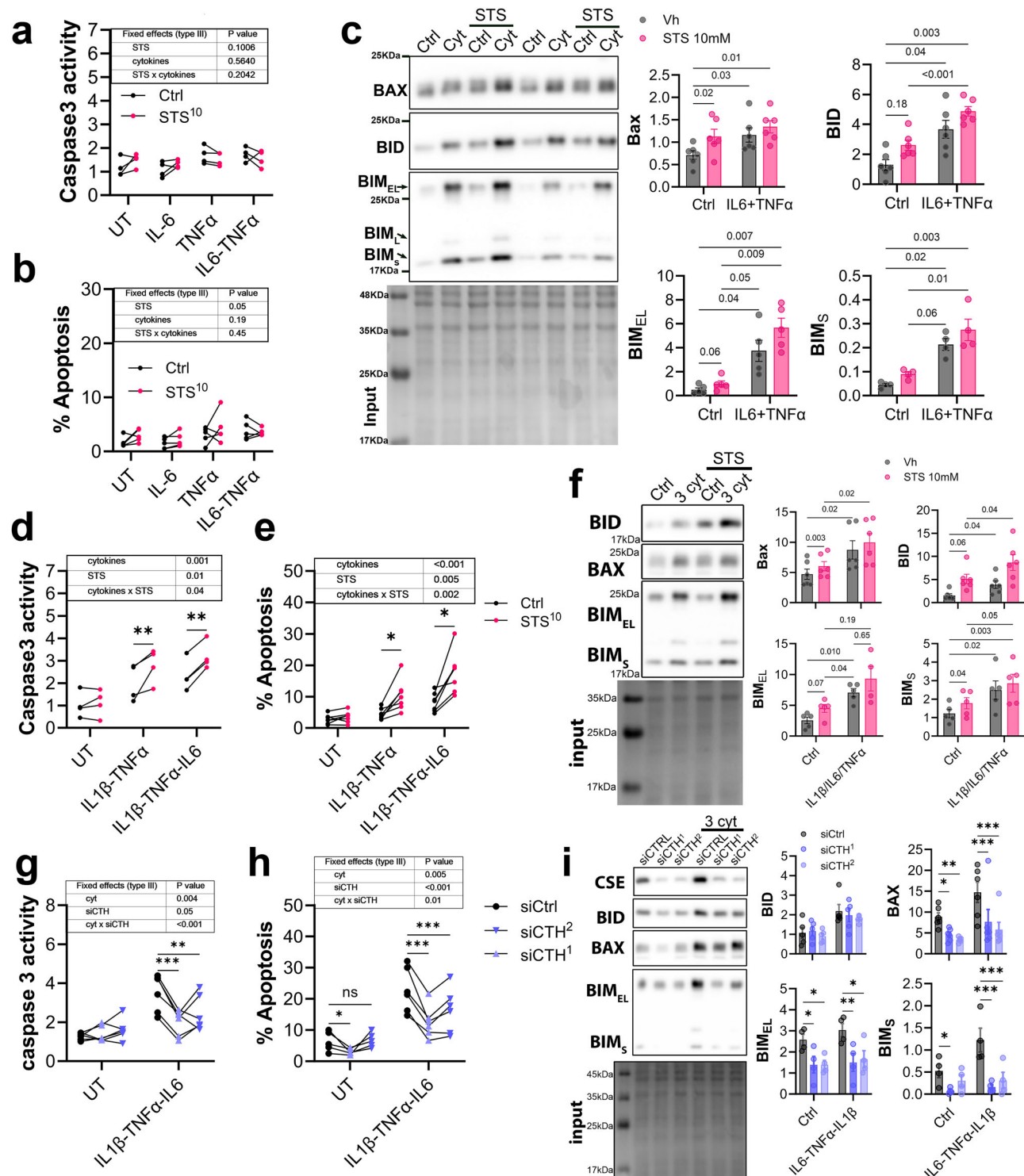

Our findings demonstrate that while hydrogen sulfide ($H_2S$) has well-established anti-inflammatory and vasorelaxant properties, its role in abdominal aortic aneurysm (AAA) progression is context-dependent and potentially harmful under normotensive conditions. The use of sodium thiosulfate (STS), a clinically relevant $H_2S$ donor, aggravated AAA growth and rupture in our murine model. This suggests a need for caution in applying STS therapeutically, particularly in populations at risk for vascular complications such as AAA. STS is increasingly used in clinical settings for conditions like calciphylaxis[82,83]. STS is also tested in several clinical trials for the treatment of ectopic calcification (NCT03639779; NCT04251832;

NCT02538939)[84,85], and as a protective agent against cisplatin-induced side effects (NCT05756660, NCT01114958, NCT05129748)[86–89]. The observed exacerbation of neutrophil-mediated extracellular matrix degradation and impact on VSMC repair underscore that STS may compromise vascular integrity in certain patient populations. This risk is particularly relevant in elderly patients or those with predisposing factors for AAA, who may already exhibit reduced VSMC repair capacity and heightened inflammation[90,91]. Thus, while STS holds promise for treating certain conditions, its use in patients with vascular pathologies should be approached with caution. Our finding calls for randomized controlled trials testing long-

**Fig. 8 | H₂S promotes cytokine-induced VSMC apoptosis.** Cleaved caspase 3 activity (**a**) and apoptosis (**b**) in vascular smooth muscle cells (VSMC) treated or not (untreated; UT) for 48 h with 10 mM sodium thiosulfate (STS) and interleukin-6 (IL-6) and/or tumor necrosis factor alpha (TNFα), as indicated. Data are the mean ± SEM of 4 and 5 independent experiments, as indicated. P values as determined by matched (paired) mixed-effects model (REML). **c** Representative Western blot (left panels) and quantitative assessment (right panels) of BAX, BID, and BIM in VSMC treated or not (Ctrl) for 24 h with IL-6 + TNFα. Data are the mean ± SEM of 4 to 6 independent experiments, as indicated. P-values as determined by matched (paired) mixed-effects model (REML), followed by Tukey's multiple comparisons tests. Cleaved caspase 3 activity (**d**) and apoptosis (**e**) in VSMC treated or not (UT) for 48 h with 10 mM STS and interleukin-1beta (IL-1β) + TNFα or IL-1β + TNFα + IL-6, as indicated. Data are the mean ± SEM of 4 to 6 independent experiments, as indicated. P values as determined by matched (paired) mixed-effects model (REML), followed by Šídák's multiple comparisons tests. **f** Representative Western blot (left panels) and quantitative assessment (right panels) of BAX, BID, and BIM in VSMC treated for 24 h with STS and IL-1β + TNFα + IL-6, as indicated. Data are the mean ± SEM of 4 to 6 independent experiments, as indicated. P values as determined by matched (paired) mixed-effects model (REML), followed by Tukey's multiple comparisons tests. Cleaved caspase 3 activity (**g**) and apoptosis (**h**) in VSMC knocked down for cystathionine gamma-lyase (CTH gene) using 2 distinct siRNAs (siCTH [1] and [2]) and treated or not (UT) for 48 h with IL-1β + TNFα or IL-1β + TNFα + IL-6, as indicated. Data are the mean ± SEM of 4 to 6 independent experiments, as indicated. P-values as determined by matched (paired) mixed-effects model (REML), followed by Dunnett's multiple comparisons tests. **i** Representative Western blot (left panels) and quantitative assessment (right panels) of BAX, BID, and BIM in VSMC knocked down for CTH using 2 distinct siRNAs (siCTH [1] and [2]) and treated for 24 h with IL-1β + TNFα + IL-6 (3 cyt), as indicated. Data are the mean ± SEM of 4 to 6 independent experiments, as indicated. P values as determined by matched (paired) mixed-effects model (REML), followed by Dunnett's multiple comparisons tests.

term administration of STS to further explore the safety and effects of STS administration on the vascular wall, particularly in patients with subclinical aortic dilations or other risk factors for AAA. Moreover, our findings suggest that future therapeutic approaches should focus on modulating neutrophil activity and preserving VSMC function, rather than broadly targeting inflammation.

## Conclusion

In summary, while the anti-hypertensive properties of H₂S offer some benefit in the context of aortic dissection and aneurysm, this study provides evidence of H₂S inefficacy in mitigating AAA growth in normotensive conditions. We further highlight a detrimental influence of H₂S on neutrophil infiltration and VSMC in this context, leading to impaired vascular remodeling and elastase-induced AAA growth. Our study challenges the prevailing assumption that H₂S donors are universally protective in vascular diseases. Previous studies have primarily focused on hypertension-driven AAA models, where the vasorelaxant properties of H₂S likely overshadow its deleterious effects. In contrast, the normotensive conditions of our study revealed a complex interplay of inflammation, VSMC dysfunction, and matrix degradation driven by H₂S, highlighting the need for model-specific considerations when evaluating therapeutic interventions. Overall, our data provided much-needed perspective on the impact of CSE and H₂S on the aortic wall in normotensive conditions. Tailored therapies that consider the specific pathophysiological context, including blood pressure and vascular integrity, will be essential to mitigate risks and harness the potential benefits of H₂S-based treatments.

## Data availability

Raw numerical data (source data) are available in Supplementary Data 1.

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

## Acknowledgements
We thank the Mouse Pathology Facility of the Faculty of Biology and Medicine, University of Lausanne, Lausanne, Switzerland, for their services in histology. We thank the Cellular Imaging Facility of the Faculty of Biology and Medicine, University of Lausanne, Lausanne, Switzerland, for their services in microscopy. The Swiss National Science Foundation to FA, grant number 310030_219997. The Union des Sociétés Suisses des Maladies Vasculaires for SD. The Fondation pour la recherche en chirurgie thoracique et vasculaire for FA and SD.

## Author contributions
C.B., D.M., F.A. and S.D. designed the study. C.B., D.M., S.U., M.L., F.C., and F.A. performed the experiments. C.B., D.M., S.U., F.C., M.L., S.D. and F.A. participated in data analysis and interpretation. C.B., D.M., S.D. and F.A. authored the paper. F.A. and S.D. obtained funding.

## Competing interests
The authors declare no competing interests.
