## [Transparent Peer Review file · Communications Medicine]

Hydrogen Sulfide Aggravates Neutrophil Infiltration, Vascular Remodeling, and Elastase-Induced Abdominal Aortic Aneurysm in Male Mice

Corresponding Author: Dr Florent Allagnat

Version 0:

Reviewer comments:

Reviewer #1

(Remarks to the Author)

In this manuscript, Dr. Bechelli and colleagues described a surprising but interesting effect of hydrogen sulfide on abdominal aortic aneurysm. The opposing effects of H₂S or Cse deficiency on macrophage and neutrophil infiltration are very interesting, however, the mechanistic investigations underlying neutrophil proliferation as well as smooth muscle cells senescence and/or apoptosis are not sufficiently developed. In addition, the scientific rigor of the work needs to be strengthened. The specific concerns or suggestions are in the following.

*1 (major) Aneurysm size (the maximum external diameter measured by ultrasound or CT or MIR or manually (animal only)) is a mostly commonly used indication of AAAs clinically and experimentally. The luminal area, although reflects the aortic expansion, can be affected by several technical parameters such as vessel constriction and tissue mounting. This reviewer highly recommends the authors to adopt the commonly used method to measure AAAs.

*2 (major). The rationale of use only male mice is not acceptable because the model used by the authors is capable of produce aneurysm dilation in both sexes.

*3 (major). Did you validate whether mice died from aortic rupture by necropsy?

*4 (major). Fig. 3, neutrophils should be detected by multiple markers. Antibodies for other neutrophil-specific markers are available.

*5 (major) Because neutrophils were thought to contribute to AAA through non-MMP9 dependent mechanism (PMID 16009808), additional evidence should be provided to support the notion that neutrophils contribute to H₂S-stimulated AAA expansion through MMP9. This is particularly important because the colocalization of IHC staining was found to be in a general area. More precise colocalization such as in situ hybridization (MMP9) with a neutrophile marker should be used. Similarly, the notion that H₂S stimulates neutrophil proliferation lacks direct proof.

*6 (major). How H₂S stimulates neutrophil accumulation but inhibits macrophage accumulation is very interesting, however it was not investigated.

7* (major). The in vitro data on the role of H₂S in SMC proliferation, senescence and apoptosis are convincing. However, whether similar changes in SMC proliferation, senescence and apoptosis take place in aneurysmal tissues need to be determined in vivo.

8 (minor). Please explain why Cse deficiency had opposing effects on elastin degradation severity and incident.

9 (minor). Saphenous veins are a reliable source of human SMCs. However, these are venous cells, and their behaviors are potentially different from aortic SMCs.

Reviewer #2

(Remarks to the Author)

On this manuscript Bechelli et al, investigate the impact of endogenously generated and exogenously administered sulfides in AAA in male mice. The authors use a special model in which they identify that sulfides have detrimental effects on VSMC survival and proliferation supporting the progression of AAA. Mechanistically the authors link such effects to cytokine mediated SMC apoptosis and elastin degradation. Overall the study is well written and the results well presented. However I have serious concerns on whether such results can mirror the physiological relevant impact of sulfides in vascular health. I am not at all convinced about first the CSE^{-/-} model used and also the STS sulfide dependent biological function vs toxicity. In particular the authors are invited to address the following concerns:

- a. CSE^{-/-} model. There is a debate on the field about global CSE^{-/-} mouse models as there are always different in phenotypes and show compensatory effects. As such the authors need to provide evidence that CSE^{-/-} used in this study have no CSE expression in the ECs, VSMCs, Fibroblasts and immune cells within the aorta. Is there any MPST or CBS compensation in those aortas? Please provide appropriate wb analysis and stainings particularly for MPST expression within the aortic cell types.
- b. The authors mention that the CSE expressing cells are infiltrating the AAA region, however this is not supported by any data. Authors should show co staining images of the key cell populations within the AAA and CSE.
- c. 10mM STS is an enormous concentration. This in combination with the fact that the authors observe reduced mitochondria respiration and VSMC death points towards a toxic effect rather than a physiological relevant effect. How are the aortic and H₂S levels in the cultured cells after STS administration compared to the physiological levels? I understand that measuring sulfides is another topic on debate on the field but there are multiple polysulfide probes that at least would give us an indication whether the authors observe such death signals and reduced mice survival due to pure toxicity. In addition if the death is cytokine mediated, do blocking Abs of those cytokines in the presence of STS rescue the death or simply what the authors describe is pure H₂S toxicity?
- d. Please assign letter on every panel within the figures and the corresponding text, so that it is easier to follow.

Reviewer #3

(Remarks to the Author)

The authors explored the role of hydrogen sulfide (H₂S) in the development of abdominal aortic aneurysm (AAA), and their experimental results showed that hydrogen sulfide (H₂S) promotes the growth of abdominal aortic aneurysm (AAA) in male mice through sodium thiosulfate (STS), increases neutrophil infiltration, and leads to the degradation of elastin, while at the same time, by inhibiting the proliferation of vascular smooth muscle cells (VSMC) and promoting cytokine-induced apoptosis, which may aggravate the pathologic process of AAA. Undoubtedly, this is an interesting attempt, but it needs minor revisions before publication.

1. It is necessary for the author to improve the quality of the manuscript, which will help the reader better understand the specific content of the study.
2. In the Introduction section, the author's description of recent advances in the study of mechanisms associated with abdominal aortic aneurysms was inadequate. This affects the fluency of reading. It is recommended that the authors make relevant additions.
3. Although the doses of STS mentioned in the study were considered safe in some studies, a systematic comparison of the effects of different doses on the progression of AAA was lacking, and it could not be confirmed whether the experimental doses used would lead to whether other vital organs were harmed or not, and it is recommended that the authors refine the comprehensive assessment of the safety and efficacy of STS.
4. In the Discussion section, it is recommended that the authors explore in more depth the anti-inflammatory mechanisms of H₂S and its dual role in AAA progression, in particular how to balance its anti-inflammatory and pro-inflammatory effects, as well as the potential modulation of immune cell interactions and communication.
5. In the Discussion section, it is recommended that the authors emphasize the clinical significance of the findings as well as the direction of future research in the Conclusion, which discusses the potential applications and risks of STS in the treatment of abdominal aortic aneurysms and other related diseases.
6. Detection of oxidative stress markers (e.g., ROS levels) and inflammatory factors (e.g., IL-6, TNF- α) is recommended to assess the regulatory role of H₂S on oxidative stress and inflammatory responses.

Version 1:

Reviewer comments:

Reviewer #1

(Remarks to the Author)

In this revised manuscript, Dr. Bechelli and colleagues made significant improvements. However, two major concerns remain.

*1 (major) Measurement of luminal area on cross sections is subject to technical influences including vessel constriction and tissue processing/mounting. This technical issue continues to be a major weakness.

*2 (major) 100% survival in the control group does not exclude the possibility that mice in the experimental group die from other causes.

Reviewer #2

(Remarks to the Author)

The authors have addressed all my concerns

Reviewer #3

(Remarks to the Author)

The team of authors answered the comments I made in full. In addition, the team of authors has made a comprehensive revision of the manuscript to address these shortcomings. I believe that this manuscript complicates the requirements for publication.

Service de Chirurgie Vasculaire
Département des Sciences Biomédicales
Rue du Bugnon 7A
1005 Lausanne

Dr Florent ALLAGNAT, PhD
Head of Research

Tél: 0216925532

florent.allagnat@chuv.ch

Below is our point-by-point answers to the reviewers' comments:

Reviewer #1 (Remarks to the Author):

In this manuscript, Dr. Bechelli and colleagues described a surprising but interesting effect of hydrogen sulfide on abdominal aortic aneurysm. The opposing effects of H₂S or Cse deficiency on macrophage and neutrophil infiltration are very interesting, however, the mechanistic investigations underlying neutrophil proliferation as well as smooth muscle cells senescence and/or apoptosis are not sufficiently developed. In addition, the scientific rigor of the work needs to be strengthened. The specific concerns or suggestions are in the following.

*1 (major) Aneurysm size (the maximum external diameter measured by ultrasound or CT or MIR or manually (animal only)) is a mostly commonly used indication of AAAs clinically and experimentally. The luminal area, although reflects the aortic expansion, can be affected by several technical parameters such as vessel constriction and tissue mounting. This reviewer highly recommends the authors to adopt the commonly used method to measure AAAs.

We agree that measuring the aortic diameter by Doppler ultrasound would be ideal. Unfortunately, the platform we have access to was not able to reliably measure the AAA diameter due to technical problems (abdominal aorta obscured by viscera). While we are optimistic about being able to do it in subsequent studies, it is not a technology we have access to at the time. The luminal area of formalin-fixed tissue is indeed an approximate proxy for AAA diameter *in vivo*. This method allowed us to maintain consistent processing across all aortic samples, ensuring accurate comparative analysis. We also performed extensive morphometric analysis, which we believe adds rigor beyond the current state-of-the-art techniques.

*2 (major). The rationale of use only male mice is not acceptable because the model used by the authors is capable of produce aneurysm dilation in both sexes.

We appreciate the reviewer's concerns, and we'd like to explain our rationale in greater detail. Focusing on male mice was based on the clinical relevance of AAA's male preponderance. While female mice could also develop AAAs, the pathophysiological mechanisms may differ, and young female mice, and even old female, may not accurately represent age-related AAA progression in women. We recognize the importance of including females in AAA research, but this study focuses on a high-risk male cohort. Investigating females with age-related conditions, such as ovariectomized or postmenopausal models, is a valuable future direction, but beyond the scope of the current study. We have clarified this rationale in the Methods section.

We agree that women's health has been overlooked for decades, and we understand the importance of promoting social justice and equality. That said, this should not be done without fully considering the specific biological, clinical, or contextual nuances. Mandating female inclusion in studies where female representation may not be biologically relevant or may confound results risks misleading or irrelevant data. In the context of age-related diseases such as cardiovascular diseases, more meaningful and applicable

results may be achieved through a more nuanced approach that respects both scientific rigor and ethical principles.

*3 (major). Did you validate whether mice died from aortic rupture by necropsy?

Unfortunately, we could not ascertain for sure that mice died of aortic rupture, but we have no reason to believe the mice died of unrelated causes as we have a near 100% survival in the control mice.

*4 (major). Fig. 3, neutrophils should be detected by multiple markers. Antibodies for other neutrophil-specific markers are available.

New co-immunostaining neutrophil elastase (NE) and MPO in mice treated with 4g/L of STS also indicate increased infiltration of MPO and NE⁺ positive neutrophils in STS-treated AAA (new Figure S5).

*5 (major) Because neutrophils were thought to contribute to AAA through non-MMP9 dependent mechanism (PMID 16009808), additional evidence should be provided to support the notion that neutrophils contribute to H₂S-stimulated AAA expansion through MMP9. This is particularly important because the colocalization of IHC staining was found to be in a general area. More precise colocalization such as in situ hybridization (MMP9) with a neutrophil marker should be used. Similarly, the notion that H₂S stimulates neutrophil proliferation lacks direct proof.

New immunostainings on mice treated with 2.5gr/L STS have been included in the new figure 3D. These results confirm that STS promotes neutrophil infiltration, even at a lower dose. Immunostaining on consecutive slides for MPO and MMP9 clearly highlighted the strong correlation between MPO and MMP9⁺ cells. New immunostaining for neutrophils elastase (NE) also confirms that neutrophils are more abundant in the aortic wall of STS-treated mice (new Figure S5). We do not claim that MMP9 is the only mediator of STS-mediated matrix degradation, but our data suggest that H₂S promotes a degradative neutrophil phenotype. On the other hand, H₂S also promoted the formation of immature neutrophil progenitor niche in the aortic wall (see detailed answer below).

We have updated our discussion to reflect the complexity of neutrophil-mediated damage in AAA, noting the formation of immature neutrophil progenitor niches.

*6 (major). How H₂S stimulates neutrophil accumulation but inhibits macrophage accumulation is very interesting, however it was not investigated.

We agree with this reviewer that this is a very interesting topic. Studies using murine models of abdominal aortic aneurysm (AAA) or thoracic aortic aneurysm (TAA) have shown increased granulopoiesis (neutrophil production) in the bone marrow and spleen (PMID: 39051127). New experiments examining circulating leukocytes 3 days post-surgery revealed that STS treatment decreased the recruitment of monocytes from the spleen and bone marrow but did not affect circulating granulocytes (new figure 4A).

Neutrophils are terminally differentiated cells that normally lack the ability to divide. Consequently, their accumulation at inflammatory sites relies on recruitment from the bloodstream rather than local proliferation. However, double immunostaining for MPO and Ki67 revealed increased "neutrophil" proliferation within the aortic wall. Recent studies, including a model of elastase-induced AAA, have identified a subpopulation of proliferative c-Kit⁺ pre-neutrophils in chronic inflammation (PMID: 39051127, 29466759). Here, we confirm the presence of Ki67⁺ neutrophils and c-Kit⁺ cells in the aortic wall, with MPO/Ki67 double-positive clusters primarily localized to the adventitia. These findings suggest the presence of immature neutrophil progenitors or a proliferative inflammatory cell subset within the aneurysmal tissue. Notably, the c-Kit⁺ and Ki67⁺ cells are concentrated in the adventitia, indicating the formation of a distinct niche of immature neutrophil progenitors at the periphery of the infiltrated zone. This aligns with emerging evidence of maladaptive responses to chronic sterile inflammation, where committed proliferative neutrophil precursors arise within inflamed tissue. Further phenotypic and functional characterization is needed to elucidate their origin, role in aneurysm pathology, and potential as therapeutic targets.

7* (major). The *in vitro* data on the role of H₂S in SMC proliferation, senescence and apoptosis are convincing. However, whether similar changes in SMC proliferation, senescence and apoptosis take place in aneurysmal tissues need to be determined *in vivo*.

New immunostainings for cleaved caspase 3 (new figure S10) were performed to assess VSMC apoptosis *in vivo*, but we observed minimal cleaved caspase 3-positive cells, likely due to rapid clearance of apoptotic cells.

New IHC-P staining of the P16^{INK4A} senescence marker were performed on AAA, revealing that 4gr/L STS stimulates VSMC senescence in the aortic wall of AAA (new Figure 7D).

We also performed double immunostaining for MPO and Ki67, showing increased neutrophil proliferation (new figure 4), but we could not ascertain significant VSMC proliferation at 14 days post-surgery. It is likely that at 14 days post-op is too late to observe substantial VSMC proliferation in that model.

8 (minor). Please explain why Cse deficiency had opposing effects on elastin degradation severity and incident.

Our working hypothesis is that CSE deficiency alters the balance between structural integrity and functional response to hemodynamic forces. The presence of increased elastin breaks in the Cse^{-/-} mice may reflect a failure in adaptive remodeling mechanisms that normally protect the aorta from mechanical stress during AAA formation. While the AAA does not expand as much due to reduced matrix degradation, the aorta may not be able to withstand the same mechanical forces as efficiently as control mice, leading to localized dissection. In other words, the lack of CSE may reduce excessive aortic dilation but might also prevent the aorta from responding appropriately to biomechanical stresses, resulting in more structural damage in the form of elastin breaks.

In other words, the lack of CSE in Cse^{-/-} mice may result in a more rigid and less responsive aorta. This could enhance mechanical stress during AAA formation, increasing wall shear stress that contribute to elastin fragmentation. A lack of H₂S-induced vasorelaxation might exacerbate the mechanical stress on the aortic wall, leading to the formation of elastin breaks and localized dissection, while the overall aneurysm size remains smaller due to the lack of excessive neutrophil infiltration and matrix degradation.

H₂S may also be involved in maintaining the balance between matrix synthesis and degradation (PMID: 34428564; PMID: 33318869). H₂S has been shown to influence the production of collagen and elastin in certain tissues. While Cse^{-/-} mice show less matrix degradation (likely due to reduced neutrophil-drive MMP9 activation), they might also have impaired matrix synthesis and repair, particularly VSMC derived matrix synthesis.

The discussion has been modified to explain this complex hypothesis better.

9 (minor). Saphenous veins are a reliable source of human SMCs. However, these are venous cells, and their behaviors are potentially different from aortic SMCs.

A new limitation has been added to mention that it would be worthwhile to test the impact of cytokines and STS on arterial VSMC.

Reviewer #2 (Remarks to the Author):

On this manuscript Bechelli et al, investigate the impact of endogenously generated and exogenously administrated sulfides in AAA in male mice. The authors use a special model in which they identify that sulfides have detrimental effects on VSMC survival and proliferation supporting the progression of AAA. Mechanistically the authors link such effects to cytokine mediated SMC apoptosis and elastin degradation. Overall the study is well written and the results well presented. However I have serious concerns on whether such results can mirror the physiological relevant impact of sulfides in vascular health. I am not at all convinced about first the CSE^{-/-} model used and the STS sulfide dependent biological function vs toxicity. In particular the authors are invited to address the following concerns:

a. CSE^{-/-} model. There is a debate on the field about global CSE^{-/-} mouse models as there are always different in phenotypes and show compensatory effects. As such the authors need to provide evidence that CSE^{-/-} used in this study have no CSE expression in the ECs, VSMCs, Fibroblasts and immune cells within the aorta. Is there any MPST or CBS compensation in those aortas? Please provide appropriate wb analysis and costainings particularly for MPST expression within the aortic cell types. The authors mention that the CSE expressing cells are infiltrating the AAA region, however this is not supported by any data. Authors should show co staining images of the key cell populations within the AAA and CSE.

We understand the concerns about the variability in phenotypes of Cse^{-/-} models. To address this, we performed new qPCR analysis, confirming that Cth is not expressed in the aorta of Cse^{-/-} mice, and that there was no dysregulation of the other sulfur-producing enzymes Cbs and Mpst (new figure S3A).

Further IHC-P experiments have been performed to study CSE expression within the aortic wall of AAA (see **Figure reviewer 1 below**). Unfortunately, we encountered a non-specific signal that could not be attributed to CSE expression. At this point, we do not have the tools to carefully assess the expression and distribution of H₂S-producing enzymes in the AAA wall. This limitation has been acknowledged in the revised manuscript. We have removed previous claims about *de novo* CSE expression in the aortic wall and edited the discussion accordingly.

Figure reviewer 1. Representative CSE immunostaining in native (sham) or 14 days post topical elastase application (AAA) sub-renal aorta from *Cse*^{+/+} (WT) or *Cse*^{-/-} male mice. Scale bars represent 80 μm. Arrowheads point to CSE-positive signal in the endothelium of native aortas. Data are representative of 4 animals per group.

10mM STS is an enormous concentration. This in combination with the fact that the authors observe reduced mitochondria respiration and VSMC death points towards a toxic effect rather than a physiological relevant effect. How are the aortic and H2S levels in the cultured cells after STS administration compared to the physiological levels? I understand that measuring sulfides is another topic on debate on the field but there are multiple polysulfide probes that at least would give us an indication whether the authors observe such death signals and reduced mice survival due to pure toxicity.

To address the concern regarding dose-dependent effects, we have now included additional data comparing 2.5g/L and 4g/L STS doses. This new data confirms that the effects observed with 4g/L STS are consistent at the lower dose. We previously showed in another model that 2.5g/L STS treatment promotes vascular repair via angiogenesis, which is incompatible with a toxic effect of STS.

Regarding STS safety, we refer the reviewer to previously published relevant data from our previous studies (supplemental Table S3 from PMID: 35334307), showing that systemic STS treatment does not cause toxicity, as evidenced by stable kidney, liver, and heart function. We are including below the relevant figures and tables from our previous studies below for convenience.

Supplementary Table S3: Blood panel

Mean (SD)	Ctrl (n=8)	STS (n=8)	p
pH	6.99 (0.07)	7.03 (0.05)	0.23
HCO ₃ (mM)	21.52 (2.48)	21.85 (1.24)	0.74
Na (mM)	150.1 (2.85)	147.5 (3.30)	0.11
K (mM)	5.6 (0.63)	4.8 (0.66)	0.02
Ca (mM)	1.39 (0.06)	1.35 (0.1)	0.33

Urea (mg/dL)	68 (8.4)	56 (10)	0.04
CK (U/L)	177.4 (47)	202.8 (50)	0.71
CK-MB (U/L)	73.8 (8.5)	86.6 (14.4)	0.39
ASAT (U/L)	75.6 (8.8)	82.4 (10.7)	0.64
ALAT (U/L)	43.9 (8.1)	39.5 (6.3)	0.69

We could not measure polysulfide production in the aortic wall due to technical limitations. However, we previously showed that STS promotes protein persulfidation *in vitro* and *in vivo* (Figure 2 from PMID:35334307). We also previously showed that STS treatment did not even double the plasma levels of thiosulfate compared to control mice (Figure 2B-C and S4 from PMID: 36262202) from about 4 to 8 μM range, while thiosulfate excretion in urine is about 800 μM . This shows that the STS treatment *in vivo* does not lead to large cytotoxic accumulation of thiosulfate *in vivo*, and that excess thiosulfate is readily excreted via the kidney.

Figure 2 from PMID: 35334307. STS increases protein persulfidation.

a) *In situ* labelling of intracellular protein persulfidation assessed by DAz-2: Cy5.5 (red), normalized to NBF-adducts fluorescence (green), in VSMCs exposed for 4 hours to NaHS (100 μM) or STS (15 mM). Representative images of 5 independent experiments. Scale bar 20 μm . **b-c)** Human TST and SUOX mRNA levels in VSMCs treated or not (Ctrl) for 24 hours with NaHS (100 μM) or STS (15 mM). **d)** Plasma polysulfides levels, as measured by the SSP4 probe, in mice treated 7 days with STS 4 g/L. Data are scatter plots with mean \pm SEM of 4 animals per group. * p < .001 as determined by bilateral unpaired *t*-test. **e)** Polysulfides levels, as measured by the SSP4 probe in liver extracts of mice treated 7 days with STS 4 g/L or NaHS 0.5g/L. Data are scatter plots with mean \pm SEM of 5 animals per group. p < .01 as determined by One-way ANOVA with post-hoc *t*-tests with Tukey's correction for multiple comparisons. **f)** Western blot analysis of Cse, Cbs and 3-Mst in Cse^{-/-} and Cse^{+/+} mice, treated or not with STS 4 g/L for 4 weeks (3 to 5 animals per group). **g)** Lead acetate assay to measure Cse-mediated H₂S production in Cse^{-/-} and Cse^{+/+} mice. Data are representative of 4 animals per group.

Figure 2 from PMID: 36262202. STS leads to enzymatic production of H₂S and increases protein persulfidation.

A) H₂S release measured by the SF₇-AM probe in HUVEC exposed for 90 min to STS. Data are mean ± SEM of 6 independent experiments. *p<.05 as determined by bilateral unpaired t-test. Scale bar 20 μm. **B)** Plasma levels of thiosulfate in mice treated or not (Ctrl) for 2 weeks with 4g/L STS. Data are mean ± SEM of 12 animals per group. p=.011 as determined by bilateral- unpaired t-test. **C)** Urine levels of thiosulfate, normalised to creatinine levels, in mice treated or not (Ctrl) for 2 weeks with 4g/L STS. Data are mean ± SEM of 6 or 7 animals per group. p=.0035 as determined by bilateral- unpaired t-test. **D)** mRNA expression in HUVEC exposed for 4h to NaHS (100μM) or STS (3mM). Data are mean ± SEM of 6 independent experiments. *p<.05, **p<.01; ***p<.001 as determined by repeated measures one-way ANOVA with Dunnett's post-hoc test. **E)** mRNA expression in gastrocnemius muscle of mice treated for 1 week with 4g/L STS. Data are mean ± SEM of 6 to 8 animals per group. *p<.05, **p<.01; ***p<.001 as determined by bilateral- unpaired t-test **F)** *In situ* labelling of intracellular protein persulfidation assessed by DAZ-2:Biotin-Streptavidin-584 (red), normalized to NBF-adducts fluorescence (green), in HUVEC exposed for 4 hours to NaHS (100 μM) or STS (3mM).

Overall, our previous findings demonstrated that thiosulfate metabolism leads to minute production of H₂S. Of note, this is in line with numerous studies showing that thiosulfate is an intermediate of sulfur metabolism releasing H₂S *in vivo* through non-enzymatic and enzymatic mechanisms (PMID: 23804280, 24962324, 26546573, 26856696).

Regarding the large concentration *in vitro*, one should consider that thiosulfate is unlikely to passively diffuse across the plasma membrane. Thiosulfate likely enters cells via specific anion transporters, such as those for sulfate or other sulfur-containing compounds (e.g., SLC13 family), where it could be metabolized into hydrogen sulfide (H₂S) or other sulfur-containing metabolites. Further studies, such as using specific inhibitors of sulfate transporters would be needed to evaluate how thiosulfate enters the cells.

In addition, if the death is cytokine mediated, do blocking Abs of those cytokines in the presence of STS rescue the death or simply what the authors describe is pure H₂S toxicity?

We thank the reviewer for this very good question. Our data suggest that STS alone *in vitro* is not cytotoxic. H₂S release/production from STS is slow and does not generate cytotoxic amount of H₂S. However, it is sufficient to induce cell cycle arrest and accelerate senescence, probably secondary to prolonged reduction of mitochondrial respiration. When exposed to pro-apoptotic cytokines, we believe the cytostatic effect of STS tips the balance toward apoptosis. While the blocking Abs experiment is a good idea, we believe it is out of the scope of this study.

Figure S10. STS is not cytotoxic at between 3-15mM in VSMC

VSMC apoptosis after 48 hours of cell culture in presence of increasing concentration of STS, as indicated. Data shown as mean \pm SEM. * $p < .05$, $p < .01$ as determined by repeated measures ordinary one-way ANOVA followed by Dunnett's multiple comparisons tests.

d. Please assign letter on every panel within the figures and the corresponding text, so that it is easier to follow.

Whenever convenient we added letters, but we did not add letter to each panel as to not overcrowd the figures when the quantifications directly match the corresponding nearby images. We believe it is clearer to put images and their respective quantifications under the same letter. The figure legends have been modified for clarity.

Reviewer #3 (Remarks to the Author):

The authors explored the role of hydrogen sulfide (H₂S) in the development of abdominal aortic aneurysm (AAA), and their experimental results showed that hydrogen sulfide (H₂S) promotes the growth of abdominal aortic aneurysm (AAA) in male mice through sodium thiosulfate (STS), increases neutrophil infiltration, and leads to the degradation of elastin, while at the same time, by inhibiting the proliferation of vascular smooth muscle cells (VSMC) and promoting cytokine-induced apoptosis, which may aggravate the pathologic process of AAA. Undoubtedly, this is an interesting attempt, but it needs minor revisions before publication.

1. It is necessary for the author to improve the quality of the manuscript, which will help the reader better understand the specific content of the study.

The manuscript has been thoroughly revised for clarity. The introduction has been revised to clarify the role of hydrogen sulfide (H₂S) and provide a more detailed explanation of the pathophysiology of AAA, particularly the contributions of neutrophils and macrophages.

2. In the Introduction section, the author's description of recent advances in the study of mechanisms associated with abdominal aortic aneurysms was inadequate. This affects the fluency of reading. It is recommended that the authors make relevant additions.

We agree with this reviewer that the introduction was a bit short. New relevant studies on the role of inflammation and senescence in the pathophysiology of the AAA are now included.

3. Although the doses of STS mentioned in the study were considered safe in some studies, a systematic comparison of the effects of different doses on the progression of AAA was lacking, and it could not be confirmed whether the experimental doses used would lead to whether other vital organs were harmed or not, and it is recommended that the authors refine the comprehensive assessment of the safety and efficacy of STS.

New immunostainings on mice treated with 2.5gr/L STS have been included in the new figure 3D. These results confirm that STS treatment promote neutrophil infiltration and immunostaining on consecutive slides of MPO and MMP9 clearly highlight the strong correlation between MPO⁺ and MMP9⁺ cells.

Regarding STS safety, we refer the reviewer to previously published relevant data from our previous studies (supplemental Table S3 from PMID: 35334307), showing that systemic STS treatment does not cause toxicity, as evidenced by stable kidney, liver, and heart function. We are including below the relevant figures and tables from our previous studies below for convenience.

Supplementary Table S3: Blood panel

Mean (SD)	Ctrl (n=8)	STS (n=8)	p
pH	6.99 (0.07)	7.03 (0.05)	0.23
HCO ₃ (mM)	21.52 (2.48)	21.85 (1.24)	0.74
Na (mM)	150.1 (2.85)	147.5 (3.30)	0.11
K (mM)	5.6 (0.63)	4.8 (0.66)	0.02
Ca (mM)	1.39 (0.06)	1.35 (0.1)	0.33
Urea (mg/dL)	68 (8.4)	56 (10)	0.04
CK (U/L)	177.4 (47)	202.8 (50)	0.71
CK-MB (U/L)	73.8 (8.5)	86.6 (14.4)	0.39
ASAT (U/L)	75.6 (8.8)	82.4 (10.7)	0.64

ALAT (U/L)	43.9 (8.1)	39.5 (6.3)	0.69
------------	------------	------------	------

Overall, our previous findings demonstrated that thiosulfate metabolism leads to minute production of H₂S. This study was already mentioned in our discussion, together with numerous other studies showing that STS is not harmful at concentration up to 2 g/Kg/day (PMID: 35334307, 23804280, 24962324).

4. In the Discussion section, it is recommended that the authors explore in more depth the anti-inflammatory mechanisms of H₂S and its dual role in AAA progression, in particular how to balance its anti-inflammatory and pro-inflammatory effects, as well as the potential modulation of immune cell interactions and communication.

The discussion has been revised to provide a more nuanced exploration of the dual roles of H₂S, emphasizing mechanisms, implications for AAA, and potential strategies for therapeutic intervention, and potential strategies to balance the dual anti-inflammatory and pro-inflammatory mechanisms of H₂S.

5. In the Discussion section, it is recommended that the authors emphasize the clinical significance of the findings as well as the direction of future research in the Conclusion, which discusses the potential applications and risks of STS in the treatment of abdominal aortic aneurysms and other related diseases. A new clinical significance section has been added in the discussion and the conclusion has been extended to integrate our findings into a broader clinical context, emphasizing the importance of cautious application of STS and suggesting directions for future research.

6. Detection of oxidative stress markers (e.g., ROS levels) and inflammatory factors (e.g., IL-6, TNF- α) is recommended to assess the regulatory role of H₂S on oxidative stress and inflammatory responses. New IHC-P staining of the P16^{INK4A} senescence marker were performed on AAA, revealing that 4gr/L STS promotes cell senescence in the aortic wall of AAA (new Figure 7D).

New experiments have been performed showing that STS tends to stimulate IL-6 expression in the aortic wall (new Fig. S6B). However, we cannot ascertain that neutrophils were the source of the staining as IL-6 signal also decorated the endothelium and media layer of the aorta.

Lauren Malave, PhD
Associate Editor,
Communications Medicine

Lausanne, Friday, June 6, 2025

Dear Dr. Malave,

This is our rebuttal letter to the comments raised by the reviewers during the second round of reviews.

Below is our detailed, point-by-point response to each reviewer.

Response to Reviewer #1

Comment 1: The authors should acknowledge that luminal area quantification from histological cross-sections may be influenced by tissue processing and may not directly reflect true aneurysm size.

Response: We fully agree with this concern. To address it, we have revised the Discussion section to explicitly mention this limitation. The following text has been added:

>“It should be noted that we were not able to directly measure the AAA size in vivo and we acknowledge that luminal area measurements from cross-sections may be influenced by tissue handling and mounting artifacts, which could introduce some variability in AAA size quantification.”

Comment 2: Mortality in the STS-treated group should be interpreted cautiously, as it is not proven to result from aneurysm rupture.

Response: We appreciate this important point and have now clarified this limitation in the manuscript. The following sentence was added to the Discussion section:

>“Moreover, although 100% survival was observed in control mice, we cannot fully exclude the possibility that mortality in the STS-treated group resulted from causes other than aneurysm rupture.”

Response to Reviewer #2

Comment: No specific additional concerns were raised by Reviewer #2

Response: We thank Reviewer #2 for their supportive comments and are glad that the previous revisions addressed their concerns.

We hope that these revisions and clarifications adequately address the reviewers' comments and improve the overall quality and transparency of our study. We appreciate the opportunity to revise our manuscript and look forward to your response.

Sincerely,

PD Dr. Florent Allagnat and PD Dr. Sébastien Déglise